# Mutations in *dnaA* and a cryptic interaction site increase drug resistance in *Mycobacterium tuberculosis*

Nathan D. Hicks[1], Samantha R. Giffen[1], Peter H. Culviner[1], Michael C. Chao[1], Charles L. Dulberger[1], Qingyun Liu[1], Sydney Stanley[1], Jessica Brown[1], Jaimie Sixsmith[1], Ian D. Wolf[1], Sarah M. Fortune[1,2,3]*

1 Department of Immunology and Infectious Diseases, Harvard T.H. Chan School of Public Health, Boston, Massachusetts, United States of America, 2 Ragon Institute of MGH, MIT, and Harvard, Cambridge, Massachusetts, United States of America, 3 Broad Institute of MIT and Harvard, Cambridge, Massachusetts, United States of America

* sfortune@hsph.harvard.edu

## Abstract

Genomic dissection of antibiotic resistance in bacterial pathogens has largely focused on genetic changes conferring growth above a single critical concentration of drug. However, reduced susceptibility to antibiotics—even below this breakpoint—is associated with poor treatment outcomes in the clinic, including in tuberculosis. Clinical strains of *Mycobacterium tuberculosis* exhibit extensive quantitative variation in antibiotic susceptibility but the genetic basis behind this spectrum of drug susceptibility remains ill-defined. Through a genome wide association study, we show that non-synonymous mutations in *dnaA*, which encodes an essential and highly conserved regulator of DNA replication, are associated with drug resistance in clinical *M. tuberculosis* strains. We demonstrate that these *dnaA* mutations specifically enhance *M. tuberculosis* survival during isoniazid treatment via reduced expression of *katG*, the activator of isoniazid. To identify DnaA interactors relevant to this phenotype, we perform the first genome-wide biochemical mapping of DnaA binding sites in mycobacteria which reveals a DnaA interaction site that is the target of recurrent mutation in clinical strains. Reconstructing clinically prevalent mutations in this DnaA interaction site reproduces the phenotypes of *dnaA* mutants, suggesting that clinical strains of *M. tuberculosis* have evolved mutations in a previously uncharacterized DnaA pathway that quantitatively increases resistance to the key first-line antibiotic isoniazid. Discovering genetic mechanisms that reduce drug susceptibility and support the evolution of high-level drug resistance will guide development of biomarkers capable of prospectively identifying patients at risk of treatment failure in the clinic.

## Author summary

Tuberculosis disease is treated with a combination of antibiotics targeting the bacterial pathogen *Mycobacterium tuberculosis (Mtb)*. In response to widespread use of antibiotics,

**Data Availability Statement:** RNA sequencing data has been submitted to the SRA database as project PRJNA667239. IDAP-seq and ChIP-seq sequencing data is submitted as project

PRJNA667510. All other data associated with this study are available in the main text or the supplementary materials.

**Funding:** This work was supported by grants from the National Institute of Allergy and Infectious Diseases https://www.niaid.nih.gov/, U19AI109755 (SMF), U19AI142793 (SMF), P01AI132130 (SMF), 5T32AI007638 (NDH), 5T32AI049928 (NDH), and 5T32AI132120 (SS). The funders had no role in study design, data collection and analysis, decision to publish, or preparation of the manuscript.

**Competing interests:** The authors have declared that no competing interests exist.

*Mtb* has evolved resistance mutations that increase the amount of antibiotic required to inhibit its growth and undermine effective treatment. The bacterial mutations that cause high-level drug resistance have largely been identified allowing for the development of rapid diagnostics. Recent studies have shown that intermediate levels of resistance can also affect patient outcomes, however, we do not yet know the range of mutations that can cause intermediate resistance. Here we utilize a genome-wide association study approach and identify that mutations in the bacterial DNA replication initiation factor *dnaA* are associated with drug resistance in clinical isolates. By generating precision *dnaA* mutant strains we identify that these mutations confer intermediate levels of resistance to the first-line drug isoniazid. We also find that mutations at a second site in the genome physically bound by *dnaA* can confer the same effect on isoniazid, likely acting through the same pathway. This study provides insight into previously unidentified clinically prevalent variants that may help explain patient outcome and guide therapy to reduce treatment failure and the subsequent evolution of high-level resistance.

## Introduction

Effective antibiotic therapy is essential to combat bacterial infections, and an accurate understanding of a pathogen's susceptibility to drug is critical for clinical management. Drug resistance is most often assayed as the ability of a strain to grow in the presence of a single 'breakpoint' concentration of a given antibiotic and is reported as a binary phenotype [1]. However, it is clear in many cases that drug susceptibility measured as the minimum inhibitory concentration (MIC) is a continuous variable [2,3]. Further, quantitative shifts in MIC below conventional breakpoint concentrations have been associated with poor clinical treatment outcome, as in vancomycin-intermediate *S. aureus* [4]. International susceptibility testing standards have now formalized the concept of strains with intermediate resistance for several bacteria where alterations in therapy are recommended [5].

In tuberculosis, an appreciable number of patients infected with phenotypically pan-susceptible *Mycobacterium tuberculosis (Mtb)* strains, as defined by breakpoint testing, fail therapy [6,7]. A recent study that sought to identify risk factors for so-called drug susceptible treatment failure found that patients who failed treatment were more likely to be infected with *M. tuberculosis* strains that had higher MICs for the first line drugs isoniazid and rifampin, although these MICs were well below the breakpoint concentration for resistance [8]. These data suggested for the first time that there are intermediate resistance phenotypes for *M. tuberculosis* that are important for therapy but provided no insight into their genetic underpinnings or molecular basis.

Bacterial population genomics has provided a powerful tool to understand and model antibiotic resistance in pathogens such as *Staphylococcus aureus* [9], *Neisseria gonorrhea* [10], and *Mtb* [11,12]. However, these models have in general focused on the prediction of breakpoint resistance to drugs [13,14]. As a result, most of the ~55,000 clinical strains of *Mtb* that have been sequenced are described with binary resistance phenotypes. Whether these data can be used to identify intermediate resistance predictors remains to be seen.

Provocatively, several recent genome-wide association studies (GWAS) of *Mtb* clinical strains have leveraged these large binary resistance cohorts and identified a constellation of variable bacterial loci associated with, but not perfectly predictive of, breakpoint resistance [15–18]. For example, we identified resistance-associated variants in the bacterial regulator *prpR* that instead confer multidrug tolerance, a form of altered susceptibility leading to

reduced bacterial killing [16]. Subsequent studies by other groups focusing on sites of selection in the *Mtb* genome identified *glpK* as an additional mediator of drug tolerance [19,20]. Based on these studies, we hypothesize that at least a subset of GWAS variants alter drug susceptibility in a way not captured by standard breakpoint testing [21], conferring instead tolerance or intermediate resistance, and these variants become associated in GWAS because they facilitate the evolution of higher level resistance [22,23].

Several technical hurdles make it difficult to predict the phenotypes of the GWAS variants identified from binary resistance cohorts. First, in GWAS studies, even known drug-specific resistance determinants are associated with resistance to multiple agents due to the high levels of co-resistance within *Mtb* clinical isolates, which are always treated with multidrug regimens. Second, identification of variants that cause intermediate resistance or drug tolerance relies on co-correlation between these primary phenotypes and the measured binary resistance phenotypes, which may obscure causal variants as the overlap in phenotypes is incomplete. Thus, in the absence of experimental investigation, the specific impact of GWAS variants and their importance in the clinic is unclear.

Here we report that in a genome wide association study of clinical *Mtb* strains, variants in the essential replication initiation factor, *dnaA*, are associated with drug resistance across two independent cohorts. We experimentally show that these variants cause intermediate resistance to the key first-line anti-tuberculosis drug, isoniazid. By mapping DnaA binding sites in the genome, we identify a novel DnaA interacting site in the *Rv0010c-Rv0011c* intergenic region which we find is also a site of selection in clinical strains. Clinically prevalent mutations in this interacting site likewise increase resistance to isoniazid. Finally, we validate these findings in an independent clinical cohort of *Mtb* strains, demonstrating that mutations in *dnaA* specifically correlate with increased isoniazid MIC. Taken together, our GWAS analysis reveals a novel pathway involving DnaA and the DnaA binding site between *Rv0010c-Rv0011c* that underlies intermediate resistance to isoniazid.

## Results

### Non-synonymous mutations in *dnaA* are correlated with resistance in *Mtb* clinical isolates

In our previous GWAS analysis utilizing strains from the Chinese national drug resistance prevalence survey [24], non-synonymous mutations in *dnaA* were associated with breakpoint resistance to isoniazid (INH) (Fig 1A, *phyOverlap* p = 0.004) [16], rifampin (RIF) (p = 0.001), and streptomycin (SM) (p = 0.004). However, *dnaA* variants were also present in 2.5% of supposedly susceptible strains (Fig 1A, inset), which is not consistent with known variants that cause breakpoint resistance [16]. To test the robustness of the association between *dnaA* variants and resistance, we examined an additional cohort of 1635 strains from Vietnam (S1 Table) [25] Phenotypic data was not available for this cohort so we first predicted the breakpoint resistance profiles for INH, RIF, and SM *in silico*. In this independent strain collection, non-synonymous *dnaA* mutations were also statistically correlated with *in silico* predicted INH resistance (Fig 1B, *phyOverlap test*, p = 0.015) and SM resistance (p = 0.003). Again, *dnaA* mutations were also present in 2.75% of apparently susceptible strains in Vietnam (Fig 1B, inset).

Among the combined 2,184 isolates analyzed, we identified a total of 59 different non-synonymous *dnaA* mutations present in 78 isolates (S1 Fig, S2 Table). Thirteen codons were targets of convergent evolution, a sign of selective evolutionary pressure [26]. Sites of selection included 10 individual nucleotide sites such as T845C, which encodes an I282T substitution and was found to evolve five times independently in five unrelated strains. DnaA is a highly-

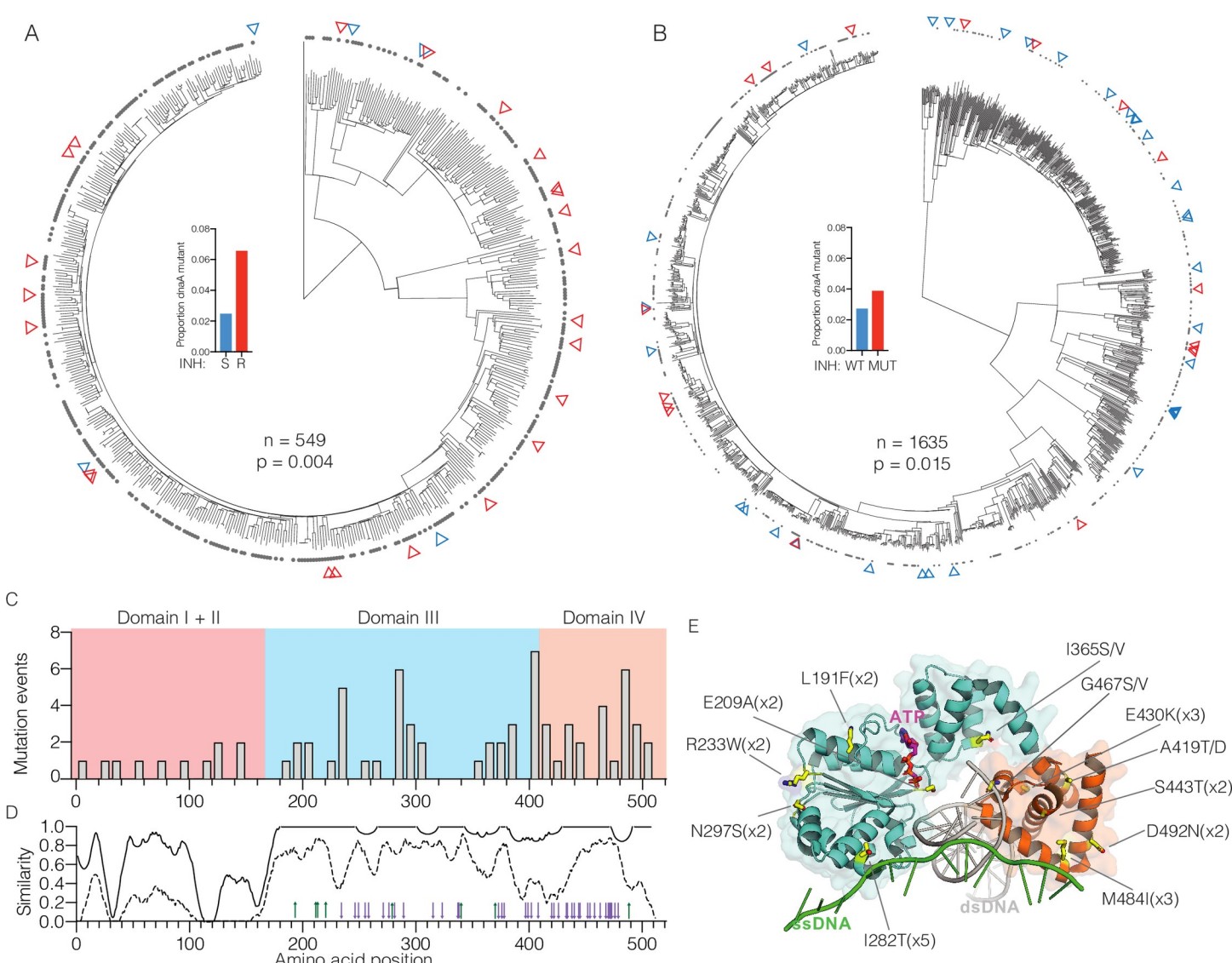

**Fig 1. Mutations in *dnaA* are clinically prevalent and associated with drug resistance.** Phylogenetic trees of (A) 549 strains from patients in China from [16]. and (B) 1635 strains from patients in Vietnam from [25]. INH resistance is marked by gray dots and non-synonymous *dnaA* mutants are indicated by red triangles for INH resistant strains and blue triangles for INH susceptible strains. Inset panels show the frequency of *dnaA* mutants among strains by INH resistance in China (S = sensitive, R = resistant), or INH *in silico* DST in Vietnam (WT = no resistance mutation, MUT = resistance mutation) (C) Distribution of mutations across the linear sequence of *dnaA* accounting for multiple evolutionary events at some residues. (D) Conservation of amino acid similarity comparing *Mycobacterial* species (solid line) or comparing *Mycobacteria* with *E. coli* and *B. subtilis* (dashed line). Characterized *dnaA* mutations in domains III and IV of *E. coli* are shown as green arrows for increased activity and purple arrows for reduced activity. (E) Composite model of *dnaA* from *A. aeolicus* and *M. tuberculosis* in complex with dsDNA (gray) and ssDNA (green). Yellow residues are found mutated in multiple unrelated isolates with the alleles noted. The R400 residue was not captured in this model.

conserved regulator of DNA replication initiation that acts through binding both DNA sequences at the origin of replication *(oriC)* and DNA replication machinery [27]. Most of the nonsynonymous *Mtb* variants occur within the regulatory ATPase domain (III) or DNA binding domain (IV) (Fig 1C). These domains exhibit high sequence conservation among both bacteria generally and mycobacteria in particular (Fig 1D). However, while 12 of the 13 convergently evolved codons are conserved between *Mtb* and the environmental mycobacterium *Mycobacterium smegmatis*, only one (G467) was absolutely conserved with *Escherichia coli* and *Bacillus subtilis* suggesting these sites may be more permissive to mutation when compared

with the rest of the domain (S2 Fig). DnaA point mutants have been extensively studied in *E. coli* in the context of cell cycle regulation (Fig 1D, arrows), however, to our knowledge no residues homologous to *Mtb* variant sites have been characterized in *E. coli* [28]. To further explore the nature of *dnaA^Mtb* mutations, we constructed a model comprised of the DNA-binding domain (domain IV) of DnaA^MTB in complex with dsDNA (PDB: 3PVP) [29] aligned with the DnaA^A. aeolicus domain III/IV complex with ssDNA (PDB: 3R8F) [30] (Fig 1E). Rather than being clustered into a single functional region or interaction surface, the sites of convergent mutation are spread throughout surfaces on both domains with several in proximity to either *ssDNA* and *dsDNA* in the composite model. As DnaA is not a known target of any clinically utilized antibiotics, these results suggest mutations are not likely to directly interfere with binding of an antibiotic and rather are selected through a downstream effect on physiology.

## *dnaA* mutations alter isoniazid susceptibility and perturb cell cycle

To validate the genomic associations and test the contribution of *dnaA* mutations to drug resistance, we constructed a panel of isogenic H37Rv strains carrying one of five different convergently evolved *dnaA* variants. For each mutation, we built three independently derived strains (except E430K which had only one representative) using oligo-mediated recombineering to introduce single, unmarked SNPs into the native copy of *dnaA* (Fig 2A). The programmatic use of prescribed first- and second-line combination therapies means that many variant genes, including *dnaA*, are associated with resistance to multiple drugs. We therefore tested the effect of *dnaA* mutations on susceptibility to a large panel of first- and second-line antibiotics with different mechanisms of action including isoniazid (INH), rifampin (RIF), streptomycin (SM), and ofloxacin (OFLX) (S3 Fig) as well as para-amino salicylic acid (PAS), sulfamethoxazole (SMX) and pyrazinamide (PZA) (S4 Fig). From this initial screening we found that all *dnaA* mutants consistently grew more robustly at the MIC of INH and conversely had decreased growth at the MIC of OFLX (S3 Fig).

To further explore these effects, we performed high-resolution competition assays using deep-sequencing to measure strain abundance during treatment of a barcoded library of mutants. At a concentration of INH near the MIC, all *dnaA* mutants outcompeted wildtype (WT) strains, becoming five to ten-fold more abundant over the course of six days of growth (Fig 2B). This was not the case for RIF (Fig 2C) or SM (Fig 2D) despite using antibiotic concentrations that inhibited growth to a similar degree (S5 Fig). Consistent with the screening assays, OFLX inhibited growth of *dnaA* mutants more than WT strains however the magnitude of the defect was substantially less than the advantage in INH (Fig 2E). In the absence of antibiotic pressure, *dnaA* mutants did not have a measurable growth advantage or defect (Figs 2F and S6), and did not exhibit differences in chromosomal initiation rate (S6 Fig), suggesting the growth advantage is due to enhanced INH resistance. *DnaA* mutants were, however, significantly longer than wildtype strains (Fig 2G), consistent with the phenotype of strains carrying hypomorphic *dnaA* alleles in other bacteria [31–35].

Recently Colangeli *et al.* demonstrated that among drug susceptible *Mtb* strains, small increases in the MIC of INH are correlated with poor treatment outcome in a continuous fashion [8]. These shifts in MIC were detected using very high-resolution MIC determination and would have been missed by standard assessments. Strains from patients with treatment failure had an average INH MIC of 0.0334 μg/ml whereas treatment successes had an average INH MIC of 0.0286 μg/ml. In contrast, the standard breakpoint resistance of INH is 0.2 μg/ml. To further define the INH susceptibility of the *dnaA* variants, we performed competition assays from 0.01–0.08 μg/ml INH at 0.01 μg/ml intervals. All *dnaA* mutants outcompeted wildtype H37Rv at a range of 0.03–0.07 μg/ml with a peak fitness difference at 0.04 μg/ml (Fig 2H).

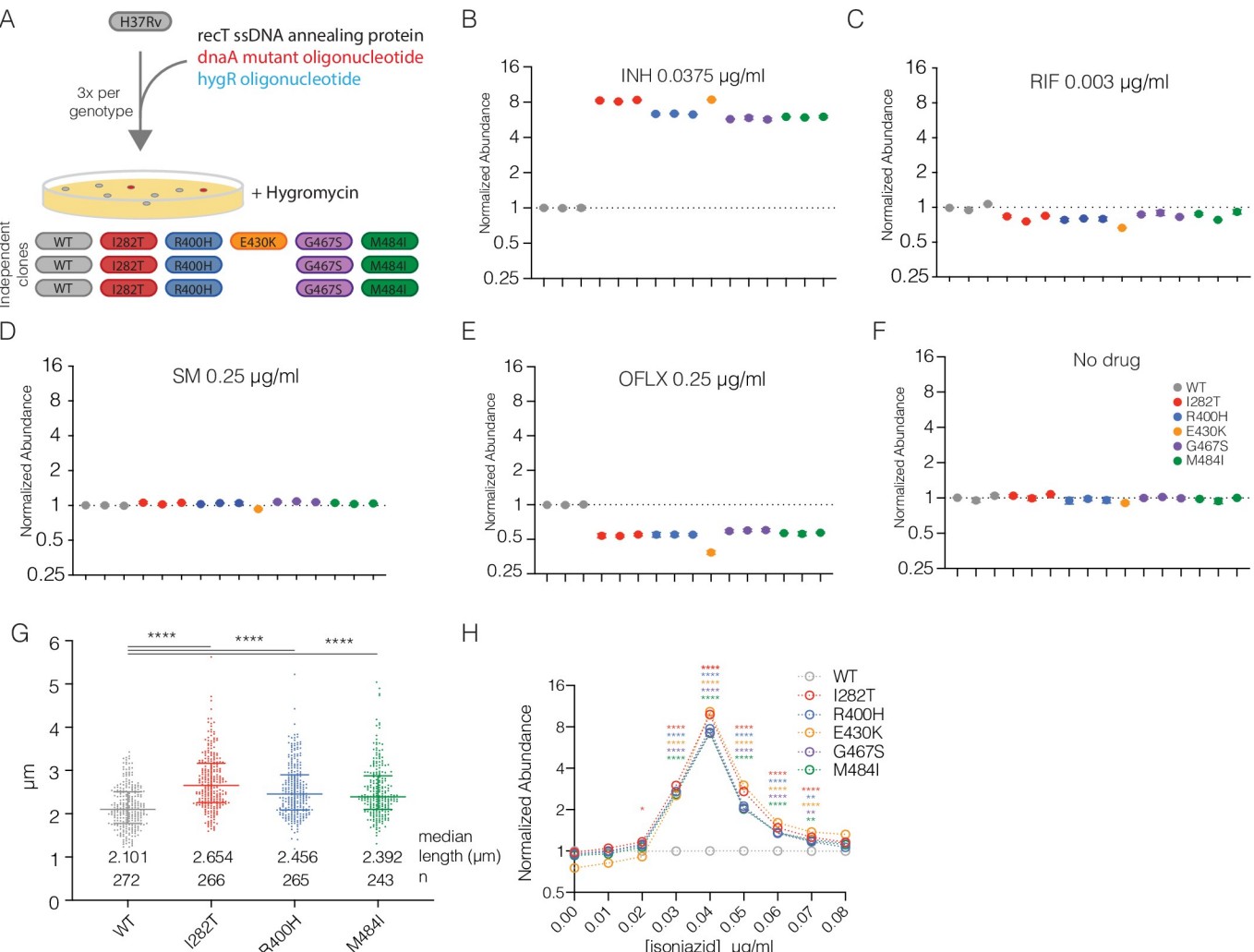

**Fig 2. Isogenic *dnaA* mutants have altered INH susceptibility and cell cycle.** (A) Schematic of the H37Rv *dnaA* mutant construction. (B-F) Competition assay with relative abundance of each strain normalized to input after 6 days of growth in the indicated antibiotic conditions. The mean and standard deviation of three technical replicates is shown with each biologically independent strain shown separately. (G) Lengths of individual bacterial cells from the indicated strains during exponential growth at 2 days post-inoculation. Median and interquartile range shown with bars. Difference among medians tested by Dunn's multiple comparison test after one-way non-parametic ANOVA. \*\*\*\* represents p < 0.0001. (H) Relative abundance of *dnaA* mutants across a range of INH concentrations. The mean of the three replicate strains (E430K n = 1) measured in triplicate cultures is shown. Error bars smaller than the symbols are omitted in D-I. Differences between each mutant and WT were tested by Dunnett's multiple comparison test after two-way ANOVA. \* <0.05, \*\* < 0.01, \*\*\*\* < 0.0001.

Together, our data show that *dnaA* mutations are sufficient to increase INH resistance. The shifts in INH MIC are on the order of those seen in patients at risk of treatment failure and confer large fitness differences in a relatively short period of drug treatment.

## Transcriptional profiling of *dnaA* mutants reveals downregulation of the isoniazid activator *katG*

To understand the basis of altered INH susceptibility in *dnaA* mutants, we performed whole-genome transcriptional profiling with RNA-seq. We compared each of three *dnaA* genotypes with WT to identify genes with significantly altered expression. We observed 113 genes with altered expression in I282T mutants, 68 genes in R400H mutants, and 63 genes for M484I mutants (Fig 3A, p-adjusted < 0.0001, S3 Table, S4 Table, S5 Table). The overlaps between

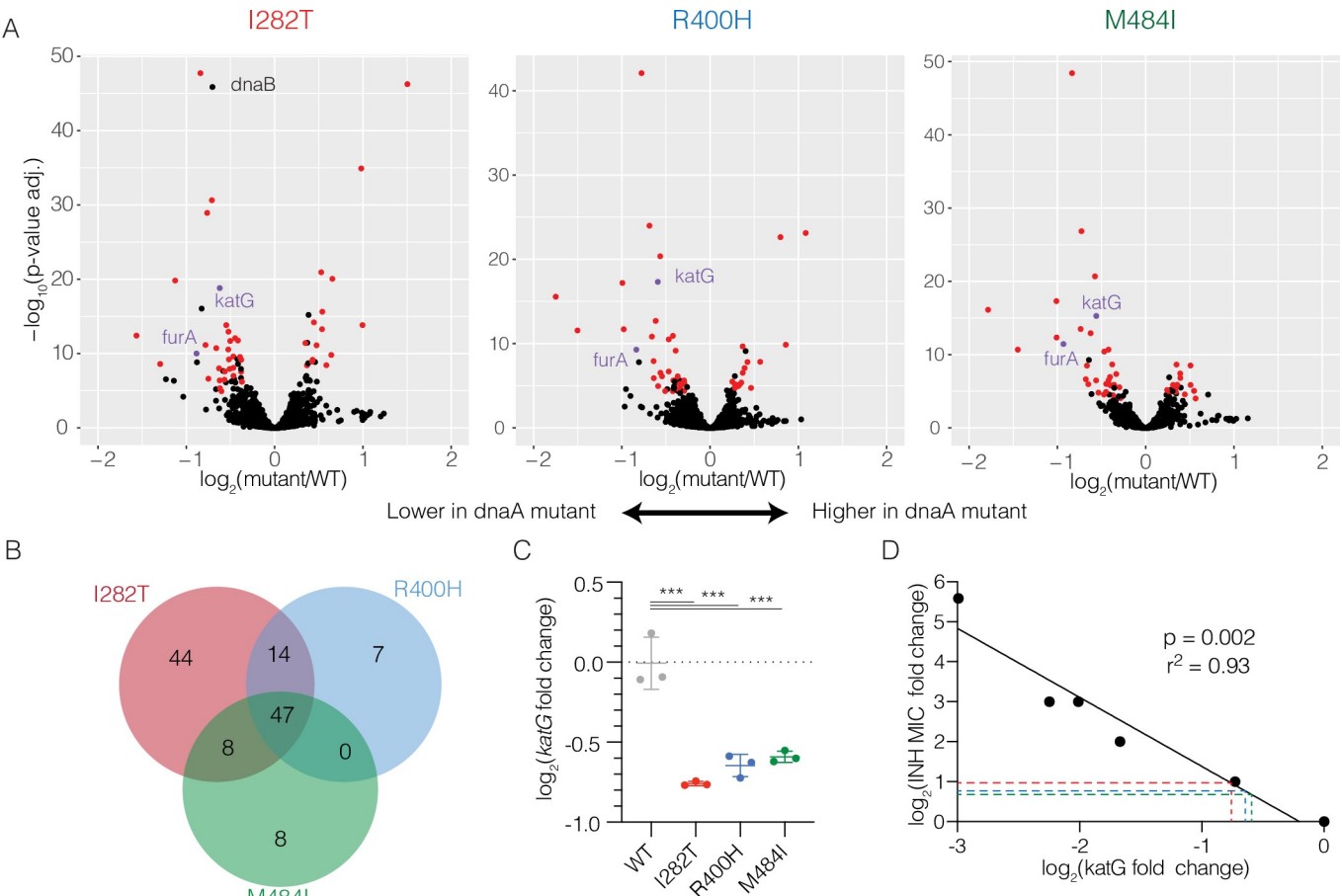

**Fig 3. Profiling of *dnaA* mutants implicates *katG* expression in INH resistance.** (A) Genome-wide differential gene expression analysis with RNA-seq. Log$_2$ fold changes are shown with increased expression in *dnaA* mutants to the right and decreased expression to the left. q-values calculated by DE-Seq analysis. Genes discussed in the text are highlighted. (B) The number of differentially expressed in each strain at q < 0.0001 compared with WT. (C) Nanostring measurement of *katG* expression normalized to composite expression of *atpH* and *sigA* using Nsolver software. Differences in mean expression tested by Holm-Sidak's multiple comparison test after one-way ANOVA. *** is p < 0.001. (D) INH MIC correlates with *katG* expression across CRISPRi mediated knockdown. Each point represents the average of two measurements from 6 different CRISPRi strains with variable levels of *katG* knockdown. Dashed lines indicate the expected change in MIC for *dnaA* mutants based on observed *katG* expression changes.

each pair of strains and among all three strains together were significantly higher than expected by chance (p < 2x10$^{-5}$ in all comparisons, see methods) with 47 genes significantly dysregulated in all three mutants (Fig 3B, S6 Table), suggesting these mutations alter cell physiology in a similar way. Notably, the expression of the bacterial activator of INH, *katG*, and its regulator *furA* was significantly lower in all three mutants (Fig 3A, purple). These differences were confirmed by Nanostring based measurements using a separate set of RNA extractions (Fig 3C). Further comparison of the differentially expressed genes with genome-wide transcriptional profiles induced by over-expression of *Mtb* transcription factors (TFs) identified several regulators with significant overlap, however none have been implicated in *furA* or *katG* regulation (S7 Table).

To determine whether the small changes in *katG* expression revealed by transcriptional profiling were sufficient to change INH susceptibility, we directly measured the quantitative relationship between *katG* expression and INH MIC. We constructed a series of CRISPRi-mediated knockdown strains in which we repressed either the *furA-katG* operon or *katG* alone to varying levels. In the absence of the knockdown, all CRISPRi strains had an identical INH

MIC. After inducing knockdown, however, we observed increased INH MIC. There was a negative correlation between measured *katG* expression and change of INH MIC (Fig 3D, β = -1.7, $r^2$ = 0.93, p = 0.002) where a two-fold reduction in *katG* expression resulted in a slightly larger than 2-fold increase in MIC, consistent with the phenotypes of the *dnaA* variants. We further found *dnaA* mutants did not have increased resistance to the second line drug ethionamide (S7 Fig) which shares the same molecular target as INH but is activated by a *katG*-independent pathway [36] or to another cell wall targeting drug, ethambutol (S7 Fig). Taken together, these data indicate that decreased *katG* expression is sufficient to explain the observed INH phenotype in *dnaA* mutants.

## Identification of a DnaA chromosomal site independently associated with drug resistance

In model bacterial species, DnaA acts directly as a transcription factor at several locations in the chromosome beyond its role at *oriC* [37]. DnaA binding sites outside of *oriC* have not been identified in mycobacteria. To understand whether changes in gene expression seen by RNA-seq are attributable to DnaA acting as a transcription factor directly at *katG* or through other regulatory pathways, we mapped the genome-wide binding sites of DnaA using *in vitro* DNA affinity purification sequencing (IDAP-seq) [38].

Recombinant 6xHIS-DnaA$^{Mtb}$ protein (S8 Fig) specifically bound *oriC*$^{Mtb}$ when compared with a control *Mtb* DNA fragment from *sigA* with a $K_d$ ~70nM (Figs 4A and S9) demonstrating the expected specificity of DnaA proteins. We then used this protein to pull down DNA sequences from randomly sheared *Mtb* genomic DNA. At 150 nM DnaA, five regions were highly enriched above background: *oriC, Rv0010c-Rv0011c, Rv0057* (upstream of *dnaB)*, *Rv3843c-RVnc0040*, and *rpmH-dnaA* (Fig 4B). These sites remained the most highly enriched at higher concentrations of protein and were insensitive to input DNA concentration (S10 Fig, S8 Table). To confirm that these sites are *bona fide* targets of DnaA *in vivo*, we additionally performed chromatin immunoprecipitation sequencing (ChIP-seq) in a strain of *M. smegmatis* in which the only copy of *dnaA*$^{Msmeg}$ has an N-terminal myc affinity tag. Remarkably, analysis of *dnaA* binding sites during exponential growth of *M. smegmatis* identified homologs of the same five regions (S11 Fig, S9 Table).

Two of these sites, *oriC and rpmH-dnaA*, are consistent with the binding profile of DnaA in other organisms where DnaA binds to *oriC* to initiate DNA replication and is autoregulated, binding to its own promoter [38,39]. DnaB is the replicative helicase and is recruited to *oriC* by DnaA. DnaA binds upstream of *dnaB*, consistent with findings in *Caulobacter crescentus* [40]. The binding sites at *Rv0010c-Rv0011c* and *Rv3843c-RVnc0040* have no obvious correspondence to binding in other bacterial species. Notably, we did not find any evidence that DnaA binds directly to the *furA-katG* genomic region suggesting the relationship between *dnaA* mutations and *katG* expression is indirect. Repeating IDAP-seq with I282T or R400H mutant protein did not identify any additional high affinity sites. In addition, the rank order enrichment of the five highly bound regions described above was similar between the mutants and wildtype protein (S8 Table), with the exception of the I282T mutant which binds less at the *dnaB* promoter and also uniquely under-expresses *dnaB* (Fig 3A).

Among the regions bound by *dnaA*, *Rv0010c-Rv0011c* is notable, as mutations, including SNPs and INDELs in this region trended toward an association with drug resistance in clinical strains from China (INH p = 0.1, SM p = 0.06) and were highly associated with *in silico* predicted SM resistance in the cohort from Vietnam (p = 0.003). *DnaA* variants were also the most strongly associated with SM resistance in this cohort. Using electromobility shift assays, we confirmed that 6xHIS-DnaA$^{Mtb}$ bound specifically to the *Rv0010c-Rv0011c* intergenic

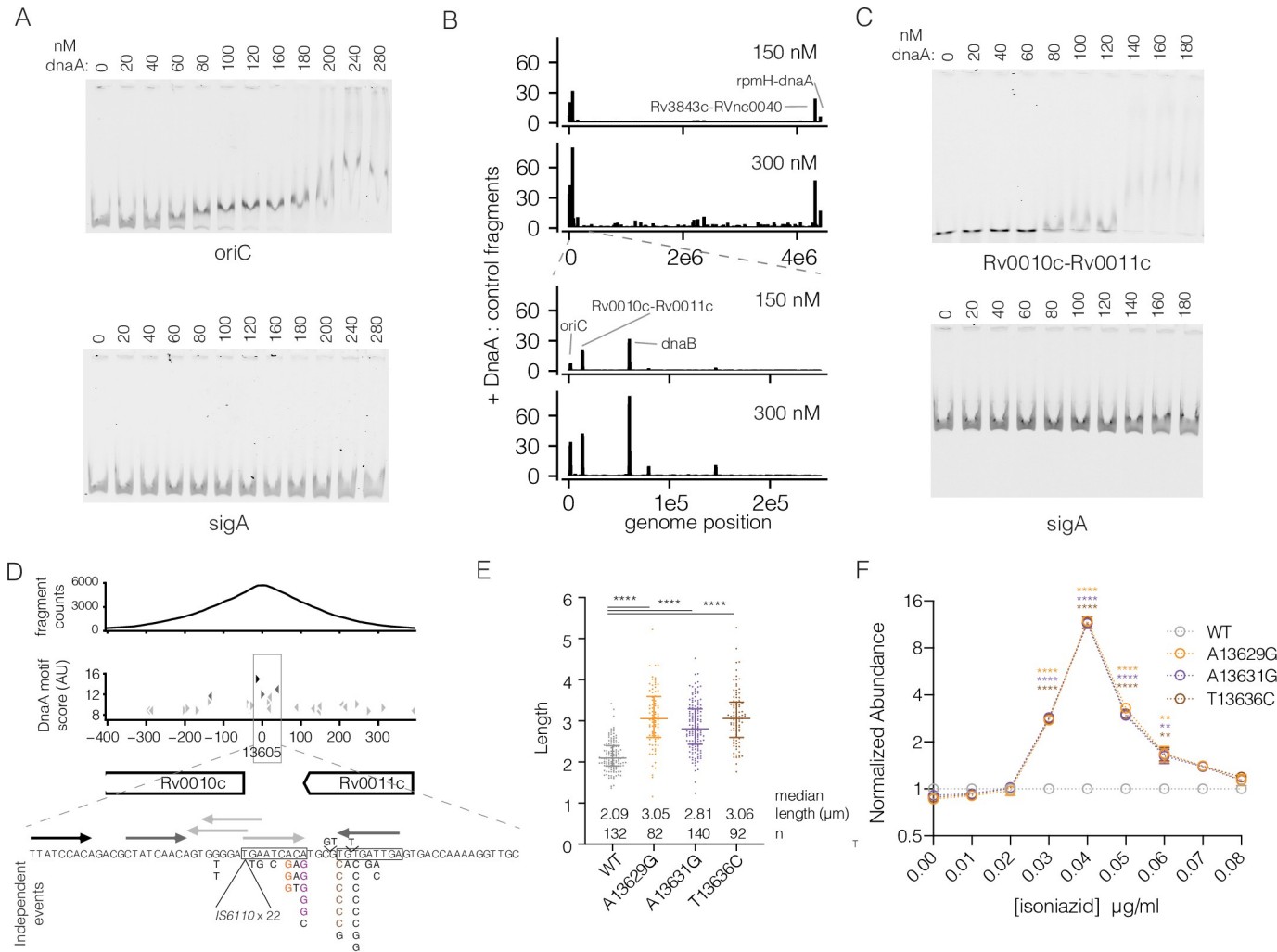

**Fig 4. DnaA binds to *Rv0010c-Rv0011c* and common mutations phenocopy *dnaA*.** (A) Electromobility-shift assay with increasing concentration of *dnaA* protein using labeled *oriC* (top panel) or *sigA* (bottom panel) DNA fragments. Both fragments were labeled with a unique dye and mixed in a single reaction. Composite images were split (B) Genome-wide binding of WT *dnaA* protein across the genome. The regions shown were split into 1000 windows and the maximum enrichment score within that window is plotted. The bottom panel shows the first 200 kb of the genome to distinguish the three closely spaced peaks. (C) Electromobility-shift assay as in panel A substituting *oriC* for a 110bp fragment of *Rv0010c-Rv0011c*. (D) Detailed view of the *Rv0010c-Rv0011c* region. Top panel: normalized fragment counts of the *dnaA* pulldown. Middle panel: Potential *dnaA* binding sites based on a position weight matrix (S10 Table) derived from biochemically identified *dnaA* binding sites from [41]. Black arrows correspond to the top scoring binding sites at 0.01%, medium gray to 0.1% and light grey to 1%. Bottom panel: sequence view of potential DnaA binding sites with clinically present mutations annotated with number of independent acquisitions. (E) Lengths of individual bacterial cells from the indicated strains during exponential growth at 2 days post-inoculation. (F) Relative abundance of *Rv0010c-Rv0011c* mutants across a range of INH concentrations. The mean of individual strains measured in triplicate cultures is shown. Error bars smaller than the symbols are omitted. Differences between each mutant and WT were tested by Dunnett's multiple comparison test after two-way ANOVA. ** $p < 0.01$, **** $p < 0.0001$.

region compared with a control sequence (Figs 4C and S9). Analysis of the DNA sequence corresponding to the center of the enrichment peak revealed a cluster of sequences resembling DnaA binding boxes based on a position weight matrix calculated from reported *dnaA* binding sites in *oriC* (S10 Table) [41].

To further assess the prevalence *Rv0010c-Rv0011c* polymorphisms in clinical strains, we re-analyzed the 2184 isolates from China and Vietnam, which identified 51 isolates (2.3%) with small SNPs and small INDELs in this region. We also identified an additional 29 isolates (1.3%) with an *IS6110* element inserted into *Rv0010c-Rv0011c*. Several sites were the targets of

convergent mutation (S11 Table), including the *IS6110* insertion which occurred 22 times across the phylogeny. All except one of these convergent loci were located within a small palindromic region located 65 bp upstream of the Rv0010c start codon (Fig 4D, boxed sequences). Several SNP sites were significantly more homoplasic than expected by chance under a null expectation that mutations could occur with equal probability at any site in the *Rv0010c-Rv0011c* intergenic region ($>$ = 6 events p = 0.00007, see methods). This palindrome is comprised of two sequences resembling *dnaA* boxes. Of the *Rv0010c-Rv0011c* mutant strains, only 2 also contained *dnaA* mutations, which is not higher than expected by chance, suggesting that mutations at either locus are not likely to be compensatory for mutations in the other (Fisher's exact test, p = 0.99).

## Mutations in the *Rv0010c-Rv0011c* DnaA binding site mediate INH resistance

To genetically test whether mutations in the *Rv0010c-Rv0011c* DnaA binding region confer phenotypes similar to *dnaA* mutations, we constructed a panel of three convergent *Rv0010c-Rv0011c* SNP variants in *Mtb* using oligo-mediated recombineering (Fig 4D, colored residues). These variants did not result in a gross defect in growth rate. However, examination of *Rv0010c-Rv0011c* mutants by microscopy again demonstrated an increase in bacterial cell length across the population (Fig 4E) which was of a similar magnitude to that of the *dnaA* variants (Fig 2C).

We then assessed the effect of the DnaA binding site variants on drug susceptibility. High-resolution competition assays demonstrated that these mutants also outcompeted wildtype at the same range of INH concentrations identified in our analysis of *dnaA* mutants (Fig 4F). Overall these data suggest that at least a subset of the prevalent *Rv0010c-Rv0011c* mutations act in the same pathway as *dnaA* mutations to modulate the cell-cycle and INH resistance.

## Mutations in *dnaA* coincide with increased INH MIC in clinical isolates and are prevalent in supposedly susceptible strains

Having uncovered evidence that *dnaA* and *Rv0010c-Rv0011c* mutations cause quantitative changes in INH susceptibility, we assessed this relationship in data from additional clinical strains. We leveraged a collection of sequenced *Mtb* isolates from Peru and the Netherlands for which MIC data are available [21]. For each non-synonymous mutant *dnaA* strain or clade, we compared the average $\log_2$-transformed MIC of the mutants with the average $\log_2$-transformed MIC of the nearest neighbor phylogenetic strain or clade without the mutation (S12 Table, S13 Table). Although multiple mutations throughout the genome distinguish each *dnaA* mutant clade and control clade, mutants had on average significantly higher resistance to INH when compared to control clades (Fig 5, median increase = 2-fold, p = 0.0038, Wilcoxon matched-pairs signed rank test two-tailed). In contrast *dnaA* mutant clades were not significantly more resistant to RIF or SM (Fig 5, RIF: median change: 0, p = 0.25; SM: median change 1.6-fold, p = 0.12). There were many fewer *Rv0010c-Rv0011c* mutants in this sample set limiting our power to assess relationships with MIC. However, like *dnaA* variants, strains with *Rv0010c-Rv0011c* mutations trended towards a higher median MIC for INH but not SM or RIF (S12 Fig), consistent with the finding that *dnaA* and *Rv0010c-Rv0011c* variants confer overlapping phenotypic effects.

In the combined cohorts of strains from China and Vietnam, *dnaA* mutations were present in ~2–3% of strains that would be treated as fully drug susceptible on the basis of currently used phenotypic or genotypic DSTs. To further understand the global prevalence of *dnaA* mutations in supposedly susceptible strains, we leveraged 51,133 whole-genome sequences

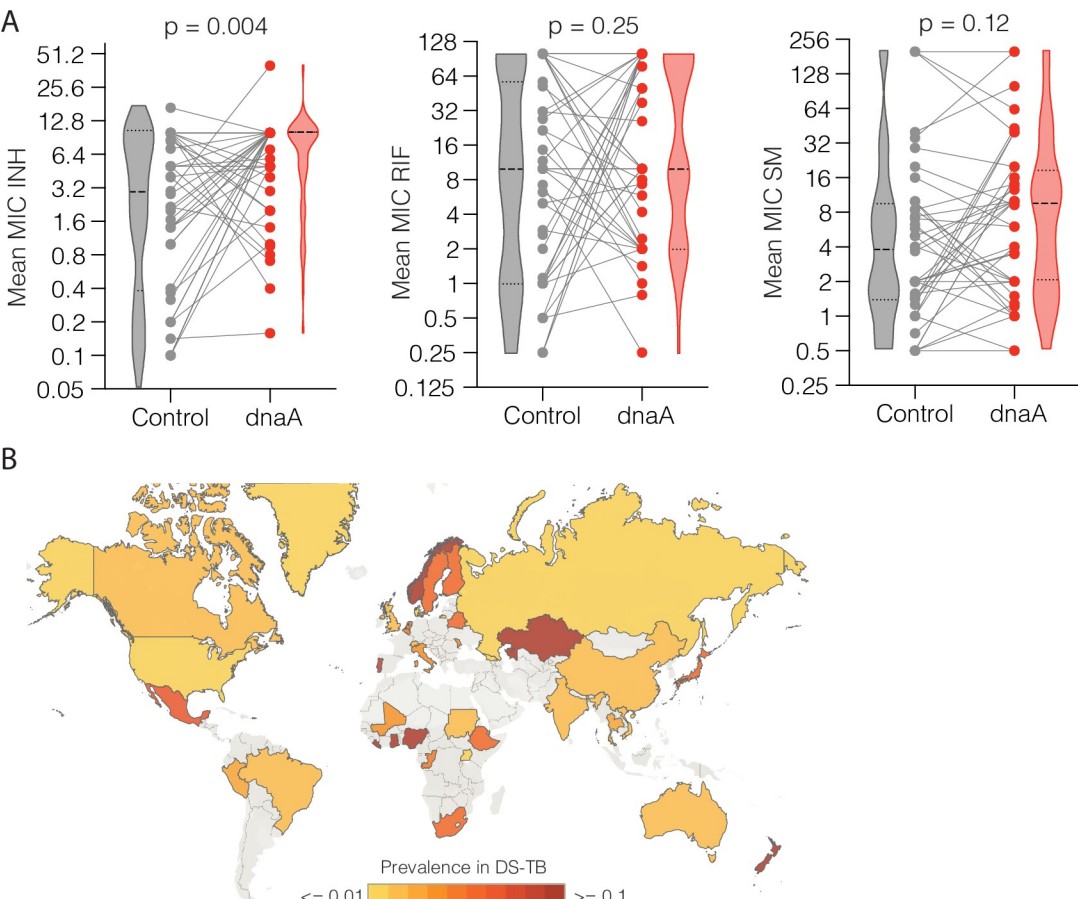

**Fig 5. *dnaA* mutations coincide with increased INH MIC and are globally prevalent.** (A) Comparison of the mean MIC of phylogenetic clades bearing *dnaA* mutations (red dots) with their nearest non-mutant neighbors (gray dots). Strong and light dashed lines indicate the median and quartile values respectively. Gray lines indicate each independent phylogenetic contrast (INH n = 43, RIF n = 43, SM n = 41). Difference in distribution tested by two-tailed Wilcoxon matched-pairs signed rank test. (B) Global prevalence of non-synonymous *dnaA* variants in presumed drug susceptible strains. The map was generated using Tableau (https://www.tableau.com/).

available on NCBI (S14 Table), of which 11,718 had a geographic label and were drug suscepti-ble by *in silico* breakpoint resistance DST as determined by the absence of known resistance conferring mutations. Globally, 3.2% of drug susceptible *Mtb* isolates contained a non-synony-mous variant in *dnaA*, where individual country prevalence varied widely from ~0.1->10% (Fig 5B, S15 Table). Thus, *dnaA* variants, which confer intermediate resistance to isoniazid at a level consistent with those found associated with treatment failure, are globally prevalent. These risk variants could be identified with expanded genotypic drug susceptibility testing.

## Discussion

In this work, we identify variants in two loci, *dnaA* and *Rv0010c-Rv0011c* that are associated with drug resistance in diverse clinical *Mtb* strains. While we initially identified *dnaA* variants through their correlation with high-level breakpoint drug resistance, *dnaA* / *Rv0010c-Rv0011c* mutations instead confer intermediate resistance to INH. These correlations suggest that *Mtb* evolution in a previously unidentified *dnaA*/ *Rv0010c-Rv0011c* pathway serves as a stepping-stone to high level drug resistance.

The emergence of intermediate resistance phenotypes is of clinical concern. Recent studies of tuberculosis patients found that strains with increasing MICs to INH and RIF were more likely to fail treatment, even though these MICs were well below the breakpoint concentration that would designate resistance [8]. INH may be particularly vulnerable to intermediate resistance because the concentration of INH in human tissues has recently been shown to fall below the breakpoint MIC during a large fraction of time during daily antibiotic administration. [42] Our data suggest that in these settings, *dnaA* and *Rv0010c-Rv0011c* mutants may outcompete wildtype strains and attenuate treatment.

These findings highlight the power of incorporating targeted experimental genetics with population genomics to expand our understanding of antibiotic resistance. Often published genomic analyses perform correlations without any experimental validation. Alleles with known phenotypes are easy to interpret even when they associate with resistance to multiple agents, and can be quickly incorporated into diagnostic algorithms. Novel alleles conferring intermediate resistance or tolerance are suggested through their imperfect, secondary co-correlation with breakpoint resistance but these associations alone are difficult to act on clinically. For example, here *dnaA* and *Rv0010c-Rv0011c* variants are associated with multiple breakpoint resistances in China and most strongly associated with SM resistance in Vietnam, where, unlike many TB programs, SM is used as a part of first line therapy. Nevertheless, our functional genetic studies found *dnaA* mutations cause intermediate resistance to INH alone and this was consistent with published clinical MIC data where we found *dnaA* mutations specifically coincide with quantitative increases in INH resistance—but not RIF or SM resistance. The strong association of *dnaA* and *Rv0010c-Rv0011c* with high-level streptomycin resistance could indicate that intermediate resistance to a particular antibiotic may help facilitate resistance to other antibiotics, as is the case for drug-specific tolerance mutations in *Staphylococcus aureus* [43]. With this study as a template, we expect that applying similar targeted genomic analysis utilizing continuous MIC data and phylogenetic contrasts, informed by experimental validation, will identify novel mutations conferring alternative phenotypes relevant to other drugs.

We find that *dnaA* mutants express less of the INH activator, *katG*, to a degree sufficient to explain their increased INH isoniazid resistance. However, further studies will be required to elucidate the full pathway connecting mutations in *dnaA* or the DnaA binding site between *Rv0010c-Rv0011c* and altered *katG* expression. Our biochemical assays did not find DnaA binding near *katG* or its known regulator *furA*, suggesting that DnaA is unlikely to be acting as a transcription factor for *katG*. A recent study showed that *katG* is variably expressed through the cell cycle [44] implying that changing progression through stages in the cell cycle could itself alter *katG* expression. Our analysis finds that cell cycle is perturbed in both *dnaA* and *Rv0010c-Rv0011c* mutants where an increase in cell length with unchanged DNA replication initiation rate implies altered coordination of the DNA replication cycle and the cell growth and division cycle. Similar phenotypes have been identified in other bacteria expressing *dnaA* variants with decreased activity [34]. However, while recent work has implicated variation in gene expression across the cell cycle with requirement of gene function [44], suggesting the catalase/peroxidase activity of KatG may be differently required across the cell cycle, further work will be necessary to understand the link between cell cycle perturbation and altered *katG* expression.

Additionally, our data suggest DnaA and the DnaA binding site at *Rv0010c-Rv0011c* form a common pathway, but the consequences of interaction also remain to be determined. It is possible *Rv0010c-Rv0011c* mutations regulate DnaA activity. Notably, neither *Rv0010c* nor *Rv0011c* expression is altered in *dnaA* mutants. In model bacteria, DnaA binding events outside of *oriC* can serve to negatively regulate DnaA by titrating it away from the *oriC* [45] or

positively regulate DnaA by facilitating nucleotide exchange [46]. We expect that further understanding of the molecular regulators of DnaA in mycobacteria, at the *Rv0010c-Rv0011c* site and through other mechanisms, may reveal additional loci contributing to variation in INH susceptibility in clinical isolates.

While additional mechanistic details remain to be dissected, the primary goal of this study was to improve our understanding of factors important to treatment of tuberculosis patients. Isoniazid is one of the most effective members of the combination therapy for active tuberculosis and used as a single agent in latent tuberculosis. Given that the absolute value of isoniazid MIC has been shown to correlate with treatment outcome, out data suggests that variants in *dnaA* and *Rv0010c-Rv0011c* may help to differentiate among patients likely to succeed in therapy versus the 5–10% of patients with drug susceptible tuberculosis who fail therapy [6,7]. These variants might also help to identify patients who are at risk of failing shortened 4-month regimens, which have been shown to successfully treat the majority of patients but have unacceptably high rates of treatment failure at the population level when only breakpoint susceptibility is used to stratify risk. [6,7,47] Given the genetic underpinnings of reduced susceptibility remain largely unknown, *dnaA* and *Rv0010c-Rv0011c* are attractive biomarkers for future analyses to assess the risk of intermediate resistance mutations leading to negative treatment outcomes.

In conclusion, we have shown that *Mtb* is independently evolving resistance to INH through mutations in a previously unknown *dnaA* / *Rv0010c-Rv0011c* pathway. These mutations are also associated with emergence of high-level resistance in diverse clinical lineages. Detecting genetic hallmarks conferring decreased susceptibility to key drugs in clinically susceptible strains could be useful for tailoring treatment regimens for patients, and at the population level may pay dividends by increasing cure rates while reducing dissemination of resistant strains.

## Methods

### Whole-genome sequence analysis

FASTQ format Illumina sequence data were downloaded from two cohorts corresponding to 1635 strains from Vietnam [25] (PRJNA355614) and 549 strains from China [16] (PRJNA268900) listed in S1 Table. Reads were aligned against the H37Rv genome reference (NC_000962.3) with the bwa mem algorithm [48]. Duplicate reads were marked with Picard tools (version 2.9.0) and variants including SNPs and small INDELs were identified with the HaplotypeCaller tool Genome Analysis Toolkit (version 3.5) [49] set to emit all sites in GVCF format on a sample by sample basis and then combined into a joint GVCF with CombineGVCFs. Genotyping was performed with the joint GVCF using the GenotypeGVCFs to create a final joint variant call file (VCF). Mutations were annotated with snpEff (version4.3) using the precompiled annotations in 'Mycobacterium_tuberculosis_h37rv'.

To test for associations between *dnaA* and drug resistance, we first constructed a whole-genome SNP phylogeny for each cohort. We selected SNP sites that were: 1) not within repetitive or known drug resistance determining regions as described in Hicks *et al.* [16], and 2) at least 90% of samples had a high quality base call defined as >10x coverage and >80% allele agreement. Individual samples with low cover (<10x) or poor allele agreement (<80% major allele) were marked as missing data. We then constructed a whole-genome phylogeny with fastTree [50]. Non-synonymous *dnaA* variants were mapped back onto the phylogenetic tree and the number of times each mutation evolved was determined by parsimony using the Fitch algorithm implemented in the *phangorn* package (version 2.5.5) [51] of R (version 3.6.1). Finally we performed the *phyOverlap* statistical test [16] to see if *dnaA* mutations were

associated with resistance. For the strains isolated in China, we used the reported phenotypes, whereas for the strains isolated in Vietnam we first performed *in silico* resistance prediction. We defined isolates as INH resistant if they had mutations in the *inhA* promoter or non-synonymous mutations in *katG* (excluding the known R463L lineage SNP) and SM resistant if they had either the K43R or K88R mutation in rpsL.

To calculate the number of times a *dnaA* variant evolved across both datasets combined, we constructed a minimal genome-wide SNP-based phylogenetic tree containing all *dnaA* mutants and their nearest phylogenetic neighbor based on the phylogenetic trees published in Holt *et al.* and Hicks *et al.* for the strains derived from Vietnam and China respectively (S1 Fig) with *Mycobacterium canetti* as the outgroup (NC_015848.1). Briefly, all selected isolates were combined into a single joint VCF as above, to identify variants genome-wide. This strategy identified 35,801 high quality SNP sites which were used to generate a phylogenetic tree with RAxML (version 8.2.11) [52] using the rapid bootstrap analysis (-f a) algorithm with 500 bootstraps using the GTRCAT model of nucleotide evolution. The phylogenetic tree was visualized and annotated using the interactive tree of life server (https://itol.embl.de/) [53].

Analysis of *Rv0010c-Rv0011c* SNPs and short INDELs was performed as with dnaA. Detection of the *IS6110* elements was performed using a custom python script which identified reads overlapping genomic positions 13600–13660 which also had a secondary alignment to another site in the genome. Each read was then searched for portions of the *IS6110* inverted 28bp element including 'AACCGCCCCGGTGAG', 'CTCACCGGGGCGGTT', 'CATGCCGG GGCGGTT' or 'AACCGCCCCGGCATG'. Samples with greater than 10 reads mapping to both the *Rv0010c-Rv0011c* IG region and containing one of these elements were considered to have an insertion present. Manual analysis of alignments confirmed that the point in reads switching from *Rv0010c-Rv0011c* to *IS6110* sequence occurred at the 'GAATC' spanning positions 13624–13628.

To statistically test whether mutations in *Rv0010c-Rv0011c* were more homoplasic than expected by chance, we compared the observed data against a null model where SNP mutation events (n = 46 events) could occur at any position in the intergenic region (n = 155 sites) randomly. We simulated the null distribution by randomly drawing 46 mutations from 155 sites with replacement and determined the maximum number of events that occurs at any site. We repeated this operation 100,000 times. The reported p-value is the fraction of times where six or more events occurred at any single site.

### Protein sequence homology and structure modeling

For the mycobacterial *dnaA* alignment, the protein sequence of *M. tuberculosis* (P9WNW3), *Mycolicibacterium smegmatis* (A0R7K1), *Mycobacterium canettii* (G0TET4), and *Mycobacterium abscessus* (B1MDH6) were used. For the cross-phylum alignment, the *M. tuberculosis*, *M. smegmatis*, *Escherichia coli* (P03004), and *Bacillus subtilis* (P05648) were used. In each case, the sequences were aligned using Clustal Omega as implemented at uniprot (https://www.uniprot.org/). Sliding window similarity shown in Fig 1D was generated by taking the alignment and converting it to a binary call where 1 = similarity in all 4 sequences and 0 = not conserved in at least 1 sequence. This was then traced in Prism (version 8) using the smoothing function with 10 neighboring sites for each point and a 6$^{th}$ order smoothing polynomial.

The model of DnaA in complex with DNA was generated in PyMol (The PyMOL Molecular Graphics System, Schrödinger, LLC.) by aligning the structure of DnaA$^{MTB}$ domain IV, in complex with dsDNA (PDB: 3PVP) [29] with the structure of DnaA$^{A. aeolicus}$ domain III/IV in complex with ssDNA (PDB: 3R8F) [30]. DnaA$^{A. aeolicus}$ domain IV was then removed for clarity. For Fig 1E, the DnaA complex model is displayed in cartoon and 60% transparent surface

representation. DnaA$^{\text{MTB}}$ domain IV is colored orange and DnaA$^{\text{A. aeolicus}}$ domain III is colored cyan. The twelve codon positions with convergent evolution are displayed as sticks and colored yellow and by element. The ATP analog is also displayed in stick representation and is colored magenta and by element. DsDNA and ssDNA are shown in cartoon representation and colored grey and green, respectively.

## *dnaA* mutant construction

*dnaA* mutants were constructed in the laboratory strain H37Rv using oligo-mediated recombineering [54]. Briefly, H37Rv was transformed with pKM427 (gift of Kennan Murphy), a chromosomal site L5 integrating vector carrying a zeocin resistance cassette and a hygromycin cassette containing a stop codon, and with pKM402 an episomal vector carrying the phage recT protein under an ATc inducible promoter to generate strain tNH95. Oligo-mediated recombineering was performed as described by Murphy *et al* [54] co-transforming a hygromycin resistance conferring with oligos conferring one of five different non-synonymous mutations in *dnaA* (S16 Table). Each *dnaA* mutation oligo was co-transformed with the hygromycin oligo into 3 separate aliquots of induced, competent tNH95 cells, recovered for 3 days in drug-free media and plated on solid media (7H10 agar, 0.5% glycerol, 10% OADC supplement, 0.05% Tween80) with hygromycin at 50 μg/ml. Colonies were then screened for *dnaA* mutation by Sanger sequencing of the *dnaA* locus (S16 Table). Three colonies from separate transformations with no *dnaA* mutation were selected to serve as the wildtype (WT) controls. All strains were expanded and then re-streaked onto solid media with hygromycin and colonies which had spontaneously cured pKM402 were identified and saved as the final strains for analysis.

Barcoded *dnaA* mutants for competition experiments were generated by transforming each *dnaA* mutant individually with a kanamycin resistant integrating vector library containing a random 7bp sequence [55]. Two colonies with a unique barcode for each strain were identified and pooled into a single mixture (referred to below as *dnaA* library) for competition assays.

## Growth curves and microscopic analysis

Indicated *dnaA* mutants were grown in liquid media (7H9, 0.2% glycerol, 10% OADC and 0.05% Tween80) to an $OD_{600}$ of ~0.8. Strains were diluted into 10ml of fresh media to a final $OD_{600}$ 0.005. Every 2 days from day 0 to day 8, samples were taken for 1) $OD_{600}$ measurement, 2) colony forming unit determination by serial dilution and plating, and 3) microscopic examination.

Cells for microscopy were fixed in 2% paraformaldehyde for 1 hour and then removed from the BSL-3 facility. Fixed cells were washed once with a quenching buffer (25mM Tris HCl pH 7.5, 50mM NaCl, 250mM glycine, 0.05% Tween80) and then resuspended in storage buffer (phosphate buffer saline, 0.05% Tween80). Cells were deposited on a 2% agarose pad and then imaged in phase-contrast with a Nikon Ti microscope using a 100x plan apo objective. Data was collected with an Andor Zyle VSC-04459 camera with dual gain ¼ with 100ms exposure using NIS-Elements AR software. Morphological analysis was performed using the MicrobeJ plugin for the FIJI implementation of imageJ. Cells were identified using a rod-shaped model and then were manually curated to select only cells which were not a part of clumps. Lengths and widths were not significantly different for strains within genotypes by non-parametric ANOVA (Kruskal-Wallis test) and so data from all three strains for each genotype were combined. Differences in length and width of cell populations was tested using the Kruskal-Wallis test followed by Dunn's multiple comparisons test. **** indicates adjusted $p < 0.0001$.

## Chromosome replication dynamics

Each of the three I282T, R400H, and M484I *dnaA* mutants and three WT strains were inoculated into liquid media at $OD_{600}$ 0.004 and allowed to grow for 3 days. $OD_{600}$ at harvest was 0.3–0.6. Eight milliliters of culture were spun down and resuspended in 1ml TE buffer. 300ul of 25:24:1 phenol:choloroform:isoamyl alcohol (PCIA) was added and cells were disrupted with bead beating. Samples were spun at 14,000RPM at 4C for 10 minutes in a tabletop centrifuge and the aqueous phase was transferred to a new tube with RNase A a final concentration of 25 µg/ml and incubated at 37C for 1 hour. 300ul of PCIA was added again and the samples were removed from the BSL-3. Phase separation was repeated and the DNA from the aqueous phase was precipitated with 1 volume of ice-cold isopropanol and 1/10 volume 3M sodium acetate pH 5.2. DNA was quantified using a Qubit and DNA sequencing libraries were prepared using the Nextera XT library preparation kit according to manufacturers instructions. Sequencing was performed on a MiSeq instrument generating 2x75bp paired end reads with a version 3, 150 cycle kit. Sequencing reads were aligned to the H37Rv reference genome and depth of coverage was calculated on a nucleotide-by-nucleotide basis using the GATK DepthOfCoverage tool. Coverage was averaged into 1000bp windows and then plotted across the length of the genome. A second-order (quadratic) line was fit to the data and the ratio of the maximum and minimum value were used as the ori/ter ratio.

## Antibiotic phenotype measurements

Alamar blue reduction assays (S3 Fig) were performed on each of the 3 strains corresponding to R400H, G467S, M484I, and WT. Strains were grown to exponential phase (~$OD_{600}$ 0.6) and then sub-cultured to 0.0015 $OD_{600}$ into 200ul of media containing antibiotics at the indicated concentrations in 96-well plates with a breathable film. Cultures were incubated for 4 days at 37C with constant shaking at which point alamarblue reagent (BioRad, BUF012) was added at 1/10 volume. The reduction of the reagent by bacterial activity was measured 24 hours later by absorbance at $OD_{570}$. Data is plotted as the mean and standard deviation among the three strains corresponding to each genotype. Differences in growth were statistically tested in Prism using Tukey's multiple comparisons test after two-way ANOVA across strains and concentrations. Adjusted p-values are represented as: $^* < 0.05$, $^{**} < 0.01$, $^{***} < 0.001$, $^{****} < 0.0001$.

Sulfamethoxazole (SMX), para-amino salicylic acid (PAS), and pyrazinamide (PZA) were tested by directly measuring the growth of strains by $OD_{600}$ in antibiotic containing media (S4 Fig). For SMX and PAS, two strains per genotype were diluted to 0.01 in duplicate wells of a 24-well tissue culture dish. with 600ul of media containing antibiotics at the indicated concentration. Culture dishes were covered with a breathable film and incubated for 6 days with constant shaking at 37C. INH at 0.04 µg/ml was included as a positive control condition. Data is plotted as the average of and standard deviation treating the mean OD of each strain as replicates for each genotype. Statistical differences were performed as above for Alamar blue.

Pyrazinamide requires acidification of the media to have efficacy against *M. tuberculosis*. The same set of strains used for SMX and PAS were inoculated into 30ml inkwell bottles containing 10ml of PZA testing media (7H9, 0.2% glycerol, 10% OADC, 0.05% tyloxopol, pH 5.8 with HCl) at $OD_{600}$ 0.01. Cultures were incubated for 6 days with constant shaking at 37C and endpoint $OD_{600}$ was measured and tested for differences among strains as above. DMSO was used to dissolve PZA so an equal volume of DMSO was used in the no drug controls.

Competition assays shown in Figs 2 and S7 were performed by inoculating the *dnaA* library at $OD_{600}$ 0.005 in 10ml of liquid media containing antibiotics at the indicated concentrations in triplicate bottles. Cultures were incubated with constant shaking for 6 days at 37C. $OD_{600}$ measurements of the bulk library were taken at 3 and 6 days post inoculation. At the end of six

days, 3ml of culture was spun down at 4000xg and the cell pellet was resuspended in 1ml of TE and gDNA was extracted as above. Illumina sequencing libraries were generated using custom primers to amplify the region containing the barcode [55] and sequenced on the Illumina MiSeq. During the library preparation process, unique molecular counters (random 9bp sequences) are added to each amplicon to correct for amplification bias and the raw abundance of each strain within each sample was calculated as the number of unique barcode/molecular counters. Normalization was performed by first dividing the raw abundance of each strain by the average abundance of the three WT strains (setting WT to 1). To account for differences in the input library, each normalized strain abundance was divided by its normalized abundance in the input library. Data is plotted as the mean and standard deviation for the strain in the three triplicate bottles. Where error bars are not apparent they are smaller than the size of the circle marking the mean.

MIC assays for *katG* depletion strains (Fig 3D) were performed as described above, however, strains were induced with 100 ng/ml anhydrotetracycline HCl for 2 days prior to initiating the experiment and for +ATc conditions, 100 ng/ml ATc was maintained in the media through the entire course of the experiment. MICs for this experiment were defined as the minimum concentration that prevented the conversion of alamarblue from blue to pink.

## Transcriptional profiling and data analysis

All three I282T, R400H, M484I, and WT strains were grown in antibiotic-free liquid media for 3 days after being inoculated at $OD_{600}$ 0.02. 7 ml of culture was spun down at 4000xg at room temperature, the supernatant was decanted and the cell pellet was resuspended in 1ml Trizol. Cells were disrupted by bead beating and then 220ul of chloroform was added. After 15 minutes of incubation samples were removed from the BSL-3 facility. RNA was isolated using the Zymo RNA miniprep kit following the manufacturers instructions and 2.2ug of total RNA were subjected to ribosomal RNA depletion using Ribominus bacteria kit (Thermofisher). RNA sequencing libraries were prepared using the Kapa RNA Hyperprep kit according to manufacturers instructions for 50ng of input mRNA (Kapa Biosystems), and sequenced on an Illumina NextSeq with single-end 75bp reads. Ribosomal depletion was not optimal, however enough reads mapped to mRNA transcripts that statistical inference could be performed for ~90% of genomic features.

RNA sequencing reads were aligned against the H37Rv reference genome using bwa mem as above. Aligned reads were assigned to genomic features using ht-seq tool 'count' [56] and reads aligning to the rRNA locus were discarded. Differential expression analysis was performed using DEseq as described in the standard vignette.

We performed a permutation test to whether the overlap among each pair of sets and the overlap of all three sets together (47 genes) was greater than expected by chance. In each permutation, for each strain, we randomly drew the number of genes identified as significant in our analysis (I282T n = 113, R400H n = 68, M484I n = 63) from the entire set of features statistically tested in our RNAseq analysis (n = 3976). We than calculated the degree of overlap among each pair of strains and among all three strains. We repeated this process 50,000 times. The p-value was the number of times we observed an overlap as high as the actual overlap in that comparison divided by 50,000.

Nanostring-based measurements of *katG* abundance was performed on RNA directly. 25ng of total RNA was hybridized to custom detection probes (S16 Table) and analyzed on a Nanostring nCounter Sprint analyzer. *katG* expression was normalized to the expression of *sigA* and *aptH* using the nSolver software provided by Nanostring. The data is plotted as the log2 transformed abundance normalized to the mean abundance of *katG* in WT strains. Differential expression was tested by Tukey's multiple comparison test after ANOVA.

### *katG* knockdown and MIC determination

KatG was knocked down using the CRISPRi system developed by Rock *et al* [57]. CRISPRi provides operonic knockdowns which reduce expression of an entire operon downstream of the guide sequence. Five guides were cloned into the pJR965 plasmid targeting *furA* or *katG*. Two guides targeted *furA*: acgtatcccggcaagccgtgt and ttggacgaggcggaggtcatc. Three guides targeted *katG*: gcccgagcaacacccacccat, agcaacacccacccattacagaaac, and cgtgggtcatatgaaataccc. These plasmids were transformed into wildtype H37Rv and selected on solid media containing kanamycin supplemented at 20µg/ml. Knockdown was induced by addition of 1mg/ml anhydrotetracycline dissolved in methanol to a final concentration of 100ng/ml culture. After 48 hours, RNA was extracted as above and strains were sub-cultured into 96-well plates for MIC determination as above. The MIC was determined as the minimum concentration to inhibit the reduction of Alamar blue to below that of the control wells. The experiment was performed twice and the data were average for Fig 3D. A linear regression was performed using Prism 8 on the $\log_2$ transformed fold-change data comparing the five CRISPRi strains against a strain with a pJR965 plasmid containing no guide sequence.

### *dnaA*$^{MTB}$ protein purification

*dnaA* was cloned into the expression vector pET28b with an N-terminal 6xHIS tag followed by a 3C prescission protease site, followed by the *dnaA*$^{MTB}$ to create pNH186. This plasmid was transformed into BL21 CodonPlus(DE3)-RP cells (Agilent, catalog 230255) to make strain *eNH1*. *eNH1* was diluted 1/100 from an overnight culture into 500ml of LB miller broth with 50 ug/ml kanamycin and 25 ug/ml chloramphenicol with shaking at 30C. At OD 0.4, protein production was induced with 0.4mM IPTG for 4.5 hours. Cells were pelleted at 5000xg and resuspended in Buffer A (50mM sodium phosphate, 5mM imidazole, 300mM sodium chloride, pH 7.2) and frozen at -80C until further processing.

Frozen pellets were thawed in warm water and sonicated on ice 1x for 45 seconds with 40% amplitude and 50% duty cycle. One EDTA-free protease inhibitor tab (Roche) and 10mg of lysozyme were added and the lysate was stirred at room temperature for 17 minutes. 2ul of benzonase was added after 10 minutes of stirring. The lysate was clarified by spinning at 35,000xg for 20 minutes at 4C. All further steps were performed at 4C. The supernatant was added to 500ul of Talon resin beads, pre-equilibrated with Buffer LG200A (45mM Hepes-KOH pH 7.6, 200mM potassium glutamate, 10mM magnesium acetate, 5mM imidazole pH 7.5, 20% sucrose) and rotated for 2 hours. The mixture of lysate and beads was added to a gravity flow column and allowed to drain. The beads were washed with 5ml of LG200A three times and then protein was eluted with 1ml fractions of LG200A with a gradient of imidazole in 10mM steps from 10mM to 100mM. After elution, EDTA was added at 0.5mM and DTT was added at 1mM final. Elution fractions were analyzed by SDS-page and the 50mM elution was snap frozen in single use aliquots in liquid nitrogen and stored at -80C for further use.

### Electromobility shift assays

Probes for electromobility shift assays (EMSA) were generated in two steps to label them with either IRDye 700 or IRDye 800 which are compatible with the Licor Odyssey system. A 550bp fragment of DNA corresponding to the *oriC* of *MTB* was amplified with primers NH739 and NH738. A 600bp fragment of DNA corresponding to a region in *sigA* wasamplified with primers NH772 and NH771. A 141bp fragment of DNA in the *Rv0010c-Rv0011c* region was amplified with NH758 and NH759. In the first PCR as short sequence handle was added and then during the second PCR, the IRDye was added to the fragment on the 5' end of the forward

primer (S16 Table). IRDye 800 was used for *oriC* or *Rv0010c-Rv0011c* and IRDye 700 was used for *sigA*.

All setup steps were carried out at 4C. *dnaA* protein was incubated with 5mM ATP for 2 hours. Binding reactions were then created using 15 ul EMSA binding buffer (40mM HEPES-KOH pH 7.6, 10mM potassium chloride, 140mM potassium glutamate, 10mM magnesium acetate, 2.5mM ATP, 0.5mM EDTA, 1mM DTT, 50ug/ml bovine serum albumin fraction V, 20% glycerol), 1 ul DNA mixture (2mM *oriC or Rv0010c-Rv0011c* probe, 2mM *sigA* probe, 100ng/ul poly-dIdC, in 10mM Tris-HCl pH 7.5), 5ul protein diluted in protein storage buffer (LG200A with 50mM imidazole, 0.5mM EDTA, and 1mM DTT). Binding was allowed to proceed for 20 minutes at 23C and then 15ul of the binding mixture was separated on a 6% DNA retardation gel (Thermofisher) with 0.5x TBE buffer at 100 volts for 2 hours in the dark at room temperature. The resulting complexes were visualized by directly imaging the gel on a Licor Odyssey imager to simultaneously detect both *oriC* and *sigA* on the 800 and 700 channels respectively.

For competitive control experiments (S9 Fig) EMSA binding buffer was reduced by 3 ul and replaced with either 3ul of 10mM Tris HCl pH8, or 3 ul of unlabeled competitor DNA dissolved in 10mM Tris HCl pH8. Specific competitor for *oriC* is the product of NH739/NH738 which produces an *oriC* sequence with no label. Similarly for *Rv0010c-Rv0011c* the specific competitor is the product of NH758/NH759 which produces the *Rv0010c-Rv0011c* fragment with no label. Unlabeled non-specific competitor for both reactions is the *sigA* fragment produced by NH772/NH771 with no label. All competitors were added at 20nM final concentration in the reaction.

### *In vitro* DNA affinity purification sequencing (IDAP-seq)

To bind ATP, wild type, I282T, or R400H DnaA with N-terminal His tags were thawed and incubated on ice for 2 hours in storage buffer (45 mM HEPES-KOH pH 7.6, 200 mM K-glutamate, 10 mM Mg-acetate, 1 mM DTT, 0.5 mM EDTA, 20% sucrose, 50 mM imidazole) with 2.5 mM freshly-dissolved ATP in 25 μL reactions. After ATP binding, reactions were mixed with 100 μL of binding buffer to achieve final buffer component concentrations of 40 mM HEPES-KOH pH 7.6, 150 mM K-glutamate, 10 mM Mg-acetate, 0.2 mM DTT, 50 μg/mL BSA, 0.1 mM EDTA, 20% glycerol, 4% sucrose, 10 mM imidazole, and 2.5 mM ATP in a 125 μL reaction volume. Binding reactions had either 4 μg or 1 μg input of Mtb genomic DNA sheared to ~400–500 bp using a Covaris E220e at 140 W, a duty factor of 10%, 200 cycles per burst, and an 80 second treatment time. DnaA concentrations in the binding reactions were either 0 nM, 150 nM, or 300 nM. After binding reactions incubated for 30 minutes at room temperature, they were mixed with a 50 μL bed volume of Talon resin (Clontech) pre-equilibrated in equilibration/wash buffer (40 mM HEPES-KOH pH 7.6, 150 mM K-glutamate, 10 mM Mg-acetate, 50 μg/mL BSA, 20% glycerol, and 2.5 mM ATP) and incubated an additional 30 minutes at room temperature with rotation. Bound DnaA-resin mixtures were then loaded onto Poly-Prep chromatography columns. Columns were washed 3 times with 1 mL of equilibration/wash buffer taking care to begin the next wash immediately after all buffer had flowed through the column. DnaA-bound genomic DNA was eluted by capping the columns and adding 500 μL of pre-warmed elution buffer (50 mM Tris pH 8, 10 mM EDTA, 1% SDS) and incubating columns on a heat block at 65°C for 15 minutes with the tops coated with parafilm to prevent evaporation. After incubation, elutions were collected and 2 additional washes of pre-warmed ChIP elution buffer was collected.

Elutions were brought to 1 mL with nuclease-free water and mixed with 100 μL of AMPure XP beads that had previously been resuspended in 1 mL custom precipitation buffer to

conserve beads (20% w/v PEG-8000, 2.5 M NaCl); apart from use of custom precipitation buffer, the standard AMPure XP purification protocol was used. Yields were estimated using the Qubit dsDNA HS kit (Thremo Fisher) and normalized to 2 ng per sample for library generation. Libraries were prepared using the NEBNext Ultra II DNA Library Prep Kit for Illumina following manufacturer guidelines for 2 ng of input DNA. Libraries were sequenced on a MiSeq instrument (Illumina) with 75 nt / 75 nt paired-end reads.

### IDAP-seq data analysis

Fragments were mapped to the Mtb H37Rv genome (NC_000962.3) using bowtie2 with the args -D 20, -I 40, -X 1000, -R 3, -N 0, -L 20, -i S,1,0.50. To generate genome-wide plots and identify peaks, we used the fragment density at each genomic location. To calculate this, a single count was added at every base a given fragment mapped to for all fragments in each experiment. Fragment densities were normalized between samples by dividing by the geometric mean of the number of fragments mapping in all samples to determine a size factor for each sample. Fragment densities at each genomic location were divided by the sample's size factor after addition of a pseudocount. To generate genome-wide plots of DnaA binding, windows of 200 nt were stepped across the genome with steps of 50 nt and the fragment density. The fragment density's sum in each window was divided by the negative control (0 mM DnaA) to determine an enrichment ratio. All windows (100 nt width, 25 nt step) with an enrichment ratio >5 in at least one library compared with 0mM DnaA is reported in S8 Table.

### ChIP-seq of myc-*dnaA*$^{Msmeg}$

To examine the binding of *dnaA* to the chromosome *in vivo* we performed chromatin immunoprecipitation sequencing in *Mycolicibacterium smegmatis (Msmeg)*. First a chromosomal site L5 integrating plasmid with nourseothricin resistance containing *dnaA*$^{Msmeg}$ under its native promoter (412bp upstream of the stard codon) was transformed into wildtype *M. smeg* strain mc$^2$155 to create the *dnaA* merodiploid strain *sNH120*. The native *dnaA* was then disrupted using double-stranded recombineering [54] to replace the region between the BstB1 sites present in *dnaA*$^{Msmeg}$ with a hygromycin resistance cassette (resulting in deletion of amino acids 160–320 and the introduction of a large intervening sequence) to create strain *sNH127*. The chromosomal plasmid at phage integration site L5 was then swapped by transforming in a L5 integrating kanamycin resistant vector, *pNH160*, containing *dnaA*$^{Msmeg}$ with an n-terminal myc affinity tag (EQKLISEEDL) between codon 1 and 2 of *dnaA*. Selecting for kanamycin resistant colonies and then screening for loss of nourseothricin resistance resulted in *sNH128* which has only a single functional copy of *dnaA* with a myc affinity tag [58]. Loss of the WT *dnaA*$^{Msmeg}$ without a n-terminal myc tag was confirmed by Sanger sequencing. In parallel, a strain identical to *sNH128* but lacking a myc-tag on *dnaA (sNH130)* was generated to serve as a control strain.

To perform ChIP-seq, both both *sNH128* and *sNH130* were grown to OD$_{600}$ 0.5–0.6 in 50ml of liquid media. Crosslinking was performed by adding formaldehyde to 1% final concentration and shaking for 30 minutes at room temperature. Crosslinking was quenched by addition of 250mM glycine and shaking for 15 minutes. Cells were pelleted at 4000xg for 10 minutes at 4C. Cells were resuspended in 500ul of Buffer P1 (20mM K-Hepes pH 7.9, 50mM potassium chloride, 10% glycerol, 1 protein inhibitor/4ml solution) and disrupted by bead beating. Lysates were clarified by spinning at 13,000 RPM for 10 minutes at 4C.

Chromatin fragmentation was performed with 130ul of lysate on a Covaris Evolution e220 with settings: Intensity 105 watts, Duty cycle 2%, 200 cycles/burst, 7 minutes. Salt concentration was adjusted to 10mM Tris HCl, 150mM NaCl, and 0.1% NP-40. 50ul of anti-c-myc

magnetic beads (Thermofisher) were washed 2x with 200ul of buffer IPP150 (10mM Tris HCl pH 8, 150mM NaCl, 0.1% NP-40) and then the sheared chromatin was added and allowed to bind at room temperature for 30 minutes with shaking. Beads were separate using a magnetic stand and the beads were washed 5x with 1ml of IPP150 and 2x with 1ml of TE buffer. The washed beads were resuspended in 150ul of elution buffer (50mM Tris HCl pH8, 10mM EDTA, 1% SDS), and incubated at 65C for 15 minutes. A second elution of 100ul was performed with 5 minutes of incubation at 65C. To reverse protein-DNA crosslinks, proteinase K was added to the elution at 1mg/ml final concentration and incubated at 37C for 1 hour and then 65C overnight. DNA was cleaned up with a PCR purification column (Qiagen) and eluted into 30ul 10mM Tris HCl.

Sequencing libraries were prepared using the NebNEXT Ultra II library preparation kit (New England Biolabs) according to manufacturers directions and sequenced on the Illumina MiSeq with paired end 75x2 bp reads. Reads were aligned against the *M. smegmatis* genome (NC_008596.1) using bwa mem as described above. ChIP peaks were identified using MACS version 2.1.2 [59] comparing the read alignments from ChIP in *sNH128* with ChIP in the control strain *sNH130*. MACS results are reported in S9 Table.

### *Rv0010c* mutant construction and phenotyping

*Rv0010c-Rv0011c* mutants were constructed as with *dnaA* mutants, however for each mutation only a single strain was constructed. Microscopic examination was performed as for *dnaA*. Competition experiments were modified to use the *Rv0010c-Rv0011c* alleles as the direct readout for sequencing. Rather than using an intergrated barcode, the *Rv0010c-Rv0011c* intergenic region was amplified with custom primers appending Illumina sequencing handles which could then be used as a substrate for library preparation as in the *dnaA* experiments (S16 Table).

### MIC comparison of Peru/Netherlands cohort

Whole-genome sequence data in SRA format was downloaded from PRJEB26000 and in assembled FASTA format from PRJNA343736. In total 1365 strains were analyzed listed in S13 Table. For whole-genome assemblies (SAMN accessions) 1.1 million 100bp paired-end sequences reads were simulated using the wgsim (https://github.com/lh3/wgsim) with the command /n/fortune_lab/nhicks/bin/wgsim-master/wgsim -e 0 -N 1100000–1 100–2 100 -r 0 FNA R1out R2out. Whole-genome variant identification and phylogenetic reconstruction was performed as described for the other two cohorts. In cases where not all isolates with a given variant were monophyletic, the variant is assumed to have evolved multiple times. For each variant-containing clade, all isolates on the neighboring branch were used for comparison. The exception was in cases where this clade contained another isolate with a variant in which case it was excluded from the control calculation. For each comparison, if no isolates in the mutant or control clade had a reported MIC, this comparison was excluded from the final analysis.

The MIC was considered the higher of two values reported if a range was given. If the MIC was reported as <X then X was used and if reported >X, then 2X was used. The MICs corresponding to each sample in mutant or control clade were log-2 transformed and then averaged. Comparison between aggregated mean MIC contrasts of mutant and control clades was performed by two-tailed Wilcoxon matched-pairs signed rank test.

### Global prevalence of *dnaA* variants

A total of 51,133 *M. tuberculosis* isolates' whole genome sequencing data were downloaded from NCBI (S14 Table), and 24,171 of them had the information of geographic origin. The fixed SNPs of all the isolates were called through a pipeline described previously [60]. The

drug-resistant profiles of each isolate were determined according to the characterized drug resistant mutations from TB Drug Resistance Mutation Database(https://www.tbdreamdb.com) and a corrected catalog of drug-resistant mutations [61]. 26,962 isolates without any known drug-resistant mutations were identified as drug-sensitive (DS) strains. To characterize the presence of *dnaA* mutants in different counties/regions, we screened for *dnaA* nonsynonymous mutations in all the DS isolates and the ratio of dnaA mutant in a country/region was given as the percentage of DS isolates with *dnaA* nonsynonymous mutations. Phylogenetic SNPs in dnaA that carried by particular sublineage/clades were excluded for the calculation of *dnaA* mutant ratio in DS strains. These phylogenetic SNPs include *dnaA* P124L (from L4.8), *dnaA* H156R (from *M. bovis*) and *dnaA* R507C (from L1.1.2). The map was generated using Tableau (https://www.tableau.com/).

## Supporting information

**S1 Fig. Whole-genome SNP phylogeny of all 78 strains containing non-synonymous dnaA mutations and their neareast, non-mutant phylogenetic neighbor.** The tree is rooted using *Mycobacterium canettii* as an outgroup. The inner ring depicts strains with non-synonymous *dnaA* mutations while each successive ring shows isolates containing a mutation shared by multiple strains. Mutations that were constructed into H37Rv are shown as stars.
(PDF)

**S2 Fig. Multispecies alignment of DnaA.** Alignment of DnaA protein sequence from Mycobacterium tuberculosis (MTB), Mycolicibacterium smegmatis (MSM), Escherichia coli (Ecoli) and Bacillus subtilis (Bsubt) with clustal omega as implemented on uniprot.org. Visualization was created in boxshade (https://embnet.vital-it.ch/software/BOX_form.html). Red indicates identity with MTB sequence. Blue indicated similarity with MTB sequence. Sites with similarity in 3 or 4 sequences are annotated in the consensus line with an upper case letter indicating 100% identity across all 4 sequences and lower case letter indicating at least one sequence without amino acid identity.
(PDF)

**S3 Fig. Drug resistance screening with alamar-blue.** Alamar blue MIC measurements of indicated WT and dnaA mutant strains against INH (A), RIF (B), SM (C), and OFLX (D). Each point represents the mean and standard deviation among three independent clones for each genotype. The dashed line in each panel indicates the concentration used for competition assays in Fig 2. Differences among Alamar blue reduction (OD570) between WT and each mutant were tested using Dunnett's multiple comparison test after 2-way ANOVA. $^*$ p < 0.05, $^{**}$ p < 0.01, $^{***}$ p < 0.001, $^{****}$ p < 0.0001.
(PDF)

**S4 Fig. Drug resistance screening by OD measured growth.** Growth of indicated strain in the presence of varying concentrations (µg/ml) of sulfamethoxazole (A), para-amino salicylic acid (B), or pyrazinamide (C) as measured by OD600 after 6 days of growth. INH 0.04ug/ml was also tested as a control. Bars represent the mean and standard deviation of two replicates. Difference in means were tested by Dunnett's multiple comparison test after two-way ANOVA. $^{****}$ p < 0.0001
(PDF)

**S5 Fig. Bulk growth measurement of the *dnaA* barcoded library.** Drug concentrations indicated are in µg/ml.
(PDF)

**S6 Fig. Growth curves and ori/ter ratio for dnaA mutants.** (A) Growth of indicated strains measured by $OD_{600}$ in standard 7H9 OADC broth conditions. (B) Growth of the same strains measured by colony forming units. (C) The ratio of ori-proximal DNA to ter-proximal DNA from rapidly growing strains. (A,B) The mean and standard deviation of the three biologically independent strains measured in technical duplicate is shown. (C) Each dot represents a biologically independent strain with the mean and standard deviation shown for a genotype. Differences tested by Holm-Sidak's multiple comparison test for indicated comparisons after one-way ANOVA.
(PDF)

**S7 Fig. Competition of dnaA mutants during treatment with partially inhibitory concentrations of ethionamide (ETH) and ethambutol (EMB).** INH is included as a positive control. (A) Final OD600 of the bulk culture after 6 days of growth. Each dot represents and independent culture with the mean and standard deviation shown. (B-F) Relative abundance of each strain normalized to input library. Dots represent mean normalized abundance and bars show the standard deviation of three replicates.
(PDF)

**S8 Fig. Purification of recombinant 6xHIS-dnaA proteins.** (A) Induction of dnaA protein production in E. coli BL21 cells for purification. Cells were boiled with Laemmli buffer, run on a 4–12% BIS-TRIS gel and visualized using SimplyBlue SafeStain (Thermofisher). (B) Purified dnaA protein elution fractions. The boxed protein purifications (50mM imidazole elution fraction) were used for experiments shown in Fig 4.
(PDF)

**S9 Fig. Competitive controls for *oriC* and *Rv0010c-Rv0011c* binding.** Specific competitors for each probe were the same DNA sequence without a label. Non-specific competitor is an unlabeled fragment of *sigA* with the same sequence as the labeled control in Fig 4A and Fig 4C. All competitors were added at 20nM final concentration during the binding reaction.'
(PDF)

**S10 Fig. Genome-wide binding of WT and mutant DnaA by IDAP-seq.** (A) Enrichment plots of the IDAP-seq analysis expressed as a maximum enrichment ratio compared with a negative control containing no DnaA protein within 1000 windows across the Mtb genome (see methods). Protein concentration, total genomic DNA input, and DnaA genotype in each reaction are indicated. (B) Data from A replotted showing only the first 250kb of the genome split into 1000 windows to allow visualization of the three independent highly enriched sites near the starting coordinate.
(PDF)

**S11 Fig. ChIP-seq of myc-DnaA in M. smegmatis.** DNA sequencing coverage plots are shown for the 5 most significantly enriched regions as determined by MACS2 analysis (see methods) as visualized in Artemis (https://www.sanger.ac.uk/science/tools/artemis). The red represents sequencing depth from a myc-tagged strain and the blue line represents sequencing depth from an untagged strain used as a negative control. The fold-enrichment values from MACS2 are annotated onto each plot. Mtb homologs of M. smegmatis genes are shown below each panel.
(PDF)

**S12 Fig. Comparison of the mean MIC of phylogenetic clades bearing any *Rv0010c-Rv0011c* mutation (red dots) with their nearest non-mutant neighbors (gray dots).** Dashed line represents the median of the dataset. Gray lines between dots indicate each independent

phylogenetic contrast. Difference in distribution tested by two-tailed Wilcoxon matched-pairs signed rank test.
(PDF)

**S1 Table. SRR data and associated phenotypes for both the China and Vietnam cohorts.**
(XLSX)

**S2 Table. All non-synonymous dnaA mutants observed in the combined 2184 strain set.**
(XLSX)

**S3 Table. DE-seq analysis of I282T vs WT.**
(XLSX)

**S4 Table. DE-seq analysis of R400H vs WT.**
(XLSX)

**S5 Table. DE-seq analysis of M484I vs WT.**
(XLSX)

**S6 Table. Genes identified as significantly dysregulated in all three mutants.**
(XLSX)

**S7 Table. Overlap of dnaA dysregulated genes with expression of genes in TF over-expression dataset.**
(XLSX)

**S8 Table. Enrichment scores for IDAP-seq across mutants, protein concentraitons, and input DNA concentrations.**
(XLSX)

**S9 Table. MACS2 identification of ChIP-peaks enriched in myc-dnaA compared with untagged control.**
(XLSX)

**S10 Table. Position weight matrix used to score potential dnaA boxes.**
(XLSX)

**S11 Table. Mutations in Rv0010c-Rv0011c among the combined 2184 strain panel.**
(XLSX)

**S12 Table. DnaA or Rv0010c-Rv0011c mutant and nearest neighbor clades used for comparisons shown in Fig 5 and S12 Fig.**
(XLSX)

**S13 Table. WGS and MIC information for strains analyzed in Fig 5.**
(XLSX)

**S14 Table. Source datasets for global analysis in Fig 5B.**
(XLSX)

**S15 Table. Raw values used to calculate prevalence of dnaA variants on a country by country basis in Fig 5B.**
(XLSX)

**S16 Table. Primers used in this study.**
(XLSX)

## Acknowledgments

We would like to thank the members of Alan Grossman's lab, especially Janet Smith, for their guidance in purifying DnaA.

## Author Contributions

**Conceptualization:** Nathan D. Hicks, Sarah M. Fortune.

**Data curation:** Nathan D. Hicks, Peter H. Culviner, Qingyun Liu.

**Formal analysis:** Nathan D. Hicks, Peter H. Culviner, Qingyun Liu.

**Funding acquisition:** Sarah M. Fortune.

**Investigation:** Nathan D. Hicks, Samantha R. Giffen, Peter H. Culviner, Charles L. Dulberger, Sydney Stanley, Jessica Brown, Jaimie Sixsmith, Ian D. Wolf.

**Methodology:** Peter H. Culviner.

**Project administration:** Sarah M. Fortune.

**Resources:** Sarah M. Fortune.

**Supervision:** Sarah M. Fortune.

**Visualization:** Charles L. Dulberger, Qingyun Liu.

**Writing – original draft:** Nathan D. Hicks.

**Writing – review & editing:** Nathan D. Hicks, Michael C. Chao, Sarah M. Fortune.

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
