## [Decision Letter · Decision Letter 0]

17 Sep 2020

Dear Dr. Fortune,

Thank you very much for submitting your manuscript "Clinical strains of Mycobacterium tuberculosis evolve increased drug resistance via mutation of a DnaA pathway" for consideration at PLOS Pathogens. As with all papers reviewed by the journal, your manuscript was reviewed by members of the editorial board and by several independent reviewers. The reviewers appreciated the attention to an important topic. Based on the reviews, we are likely to accept this manuscript for publication, providing that you modify the manuscript according to the review recommendations.

All reviewers had comments that can likely be readily addressed. Some involved including statistical analyses of statements in the manuscript and others may involve additional explanation and discussion. Note that the major comments for erviewer 2 are included in the overall summary. The EMSA assay requires a cold competitive control.

Sincerely,

Helena Ingrid Boshoff

Associate Editor

PLOS Pathogens

Kasturi Haldar

Section Editor

PLOS Pathogens

Kasturi Haldar

Editor-in-Chief

PLOS Pathogens

orcid.org/0000-0001-5065-158X

Michael Malim

Editor-in-Chief

PLOS Pathogens

orcid.org/0000-0002-7699-2064

All reviewers had comments that can likely be readily addressed. Some involved including statistical analyses of statements in the manuscript and others may involve additional explanation and discussion. Note that the major comments for erviewer 2 are included in the overall summary. The EMSA assay requires a cold competitive control.

Reviewer Comments (if any, and for reference):

Reviewer's Responses to Questions

**Part I - Summary**

Reviewer #1: Determining the genetic basis for drug tolerance and resistance is necessary to developing shorter, more effective regimens for microbial diseases. In Mycobacterium tuberculosis, whilst the genetic basis for most common forms of drug resistance has been described, there is recent emerging evidence for a greater complexity in the manifestation of phenotypic drug resistance, rather than just the binary description of drug susceptibility adopted in clinical diagnosis and management. Intermediate levels of resistance are known to play an important role in treatment outcome and the genetic basis of these effects still require study. In this manuscript, Hicks and colleagues assess resistance to isoniazid (INH) using a combination of GWAS and modern genetic approaches. The authors show that in two cohorts, one from China and a second from Vietnam, non-synonymous mutations in dnaA, involved in initiation of DNA replication, were associated with INH resistance. Most mutations appear in the ATPase or DNA binding domains of the protein and as this protein is not known to be the target of any antibiotics, the authors predict that any association with drug resistance most likely arises through downstream effects on metabolism. To investigate, the authors construct mutants of H37Rv, carrying one of five possible mutations in DnaA and demonstrate that these mutants grow better than the parental strain in INH and other antibiotics tested. In competition experiments, all DnaA mutants outcompeted wildtype, suggesting that mutations in this protein are sufficient to change susceptibility to INH. Next, the authors demonstrate that expression of katG and furA was lower in DnaA mutants, an effect which the authors recapitulate using CRISPRi-mediated knockdown to demonstrate that these changes in gene expression correlate with INH susceptibility. Using recombinant DnaA (either wildtype or mutated versions) the authors identified DnaA binding sites in the genome but could not locate binding sites upstream of katG region. Instead, DnaA bound to the Rv0010-Rv0011 region, which was associated with drug resistance in Chinese clinical strains. Further comparative genomics revealed that many strains carried mutations or insertions in this region and the authors construct three such mutations, which result in longer cell lengths and increased resistance to INH, as described for their earlier DnaA mutants. Finally, the authors investigate the prevalence of DnaA mutations in different clades of M. tuberculosis and confirm that these are associated with higher resistance to INH. Collectively, the authors demonstrate that DnaA mutations affect susceptibility of M. tuberculosis to an important frontline drug used to treat tuberculosis. The mechanism through which this occurs appears to relate to lower levels of katG, but how DnaA exerts these effects is unknown.

Reviewer #2: This work links mutations at dnaA and its binding site with increases in MIC to isoniazid. Increases in MIC that do not meet breakpoint thresholds are increasingly recognized as clinically important and the study directly addresses this important topic. On the whole I think the study is interesting and well done and I have few suggestions for improvement. My specific comments are as follows:

1. The authors state in the abstract that “a DnaA interaction site… is under selection in clinical strains.” Homoplastic mutations are reported at this site (lines 300-310), but I do not see a formal assessment of whether more homoplasies are observed that is expected by chance. This should be done. In addition, it would in my opinion be more accurate to say the sites were subject to repeated mutation or to specifically describe the type of selection the authors speculate is at work here.

2. It’s not clear how the GWAS was done on the Vietnamese sample as the resistance phenotypes are described as being predicted in silico. In that case, the analysis would be an assessment of co-occurrence of dnaA mutations and known INH conferring mutations. Please clarify.

3. Line 201: The overlap among genes differentially expressed in dnaA mutants is described as “relatively high”. Please provide a formal statistical assessment of whether or not there is more overlap than is expected by chance.

4. Geographic patterns of dnaA mutations are shown in Figure 5 that reveal marked differences in regional prevalence. What do the authors make of these patterns? Is there any relationship between dnaA mutations and prevalence of MDR-TB? What do the authors make of the fact that dnaA mutations appear uncommon in Russia, where MDR-TB is common?

5. It’s not clear how the authors called short INDELs in dnaA (and Rv0010c-0011c).

Reviewer #3: This article by Hicks, Fortune and colleagues extends previous work by the same authors in which a GWAS-type analysis is used to identify previously unknown polymorphisms that are distinct from the well-described genetic resistance mechanisms but which nevertheless subvert antibiotic efficacy. In this study, the authors present evidence that non-synonymous mutations in dnaA and a putative DnaA-binding region (located intergenically in the Rv0010c-Rv0011c) locus are associated with low-level resistance of Mycobacterium tuberculosis (MTB) to the frontline anti-tuberculosis drug, isoniazid. This work is important in expanding the set of novel resistance mechanisms (recent work by the authors and others have implicated prpR and glpK in analogous “non-classical” resistance mechanisms) that might contribute to the rapid emergence of high-level antibiotic resistance in MTB, however the concerns detailed below should be addressed.

**Part II – Major Issues: Key Experiments Required for Acceptance**

Reviewer #1: The study is executed well and reported clearly. There are no major concerns

Reviewer #2: No major experiments. See above for minor additions.

Reviewer #3: 1. Figures 1C and 1D present analyses of the locations of the different non-synonymous SNPs identified in dnaA. As the authors point out, the different domains (especially domains III and IV) are highly conserved: however, the sequence-level analyses in panel D appear to indicate that most of the mutations found in the MTB isolates occur in regions that are least conserved when comparing mycobacteria with E. coli and B. subtilis (as inferred from the dashed lines).

2. In Figure 1 and throughout the manuscript: it is critical for the interpretation of the results that the authors indicate clearly whether all analyses are limited solely to non-synonymous SNPs and mutations that alter (or disrupt) the amino acid sequence; that is, all synonymous mutations are excluded. This is especially the case given that all functional validations presented in the manuscript are limited to non-synonymous mutations.

3. The authors note (Lines 291-5) that dnaA and Rv0010c-Rv0011c mutations were associated with streptomycin resistance in the Chinese and Vietnamese cohorts.

a. How is this observation reconciled with the apparent absence of a competitive growth advantage (or alteration of streptomycin MIC) in the engineered dnaA mutants?

b. Similarly, why was the impact of Rv0010c-Rv0011c on streptomycin MIC and competitive growth advantage not determined (in Fig. 4)?

c. Finally, given the apparent association in clinical isolates of Rv0010c-Rv0011c mutations with streptomycin resistance, what is the rationale for looking at INH susceptibility and, moreover, how is altered INH explained?

**Part III – Minor Issues: Editorial and Data Presentation Modifications**

Reviewer #1: 1. EMSA assays are missing cold-competitive controls. These are standard in the field and should be included, especially with transcriptional regulator-type proteins.

2. Line 184 – there is a word missing, perhaps should read, “…correlated with poor treatment…”

Reviewer #2: Line 17 should be “is associated with poor”

Line 149 missing “of” between “panel” and “first”

Line 375: remove “that”

Line 490- multiple mis-spellings of Mycobacterium

Reviewer #3: 1. The title claims that resistance is acquired via “mutation of a DnaA pathway”. The meaning of this is unclear/inaccurate (what’s a “DnaA pathway”?) and the authors should consider a more informative alternative.

2. L17: “is associated”

3. L107: The authors state that “thirteen codons were targets of convergent evolution” however only 12 appear to be presented in Fig S1 and Table S2

4. L108: the T845C mutation is indicated a synonymous SNP (“T845T”) in Table S2.

5. L120 (and elsewhere): The strains are not “Chinese” or “Vietnamese”: they are M. tuberculosis bacilli isolated from individuals in those countries. COVID-19 has provided a stark example of the dangers inherent in labelling microbes with potentially demonizing nationalities.

6. The value of the modelling data presented in Fig. 1E is unclear: it seems doubtful that anyone would have expected the identified mutations to impact/alter antibiotic (INH) binding. Did the authors learn nothing from this analysis that might better inform potential explanations relating to the impact on ssDNA versus dsDNA binding?

7. In testing the first-line anti-TB agents against the engineered dnaA mutants (Fig. 2 and Fig. S2), why was ethambutol not included in the panel of antibiotics assayed?

8. Fig. 2: It would be more helpful if the antibiotic concentrations used in the competition assays were presented as a function of their MICs (e.g. 0.5X; 0.75X, etc.)

9. L184: “correlated with poor”

10. L303: Here and elsewhere “IS6110” (italics)

11. Figure 5: What is a "nearest non-mutant neighbor"; specifically, how “isogenic” are these strains – that is, are genetic differences limited primarily to dnaA SNPs or are there many others? Again, as noted in comment 2 (above), are only non-synonymous mutations included in this analysis?

12. L357: It is not clear what is meant by “in silico breakpoint resistance” (L357); the authors should provide more detail about how this was determined.

13. L406-8: This statement is no longer accurate, and the data presented in this manuscript should be discussed in the context of recent work from Sassetti and colleagues (doi: 10.1016/j.cub.2020.07.070).

14. Supplementary Figure 2: no labels are provided indicating which plot relates to which mutant.

PLOS authors have the option to publish the peer review history of their article (what does this mean?). If published, this will include your full peer review and any attached files.

Reviewer #1: No

Reviewer #2: No

Reviewer #3: No
---

## [Editor Report · Decision Letter 1]

9 Oct 2020

Dear Dr. Fortune,

We are pleased to inform you that your manuscript 'Mutations in dnaA and a cryptic interaction site increase drug resistance in Mycobacterium tuberculosis' has been provisionally accepted for publication in PLOS Pathogens.

Best regards,

Helena Ingrid Boshoff

Associate Editor

PLOS Pathogens

Kasturi Haldar

Section Editor

PLOS Pathogens

Kasturi Haldar

Editor-in-Chief

PLOS Pathogens

orcid.org/0000-0001-5065-158X

Michael Malim

Editor-in-Chief

PLOS Pathogens

orcid.org/0000-0002-7699-2064

The authors have addressed the reviewers' concerns. The work demonstrates that non-synonymous mutations in dnaA confer intermediate levels of drug resistance in clinical strains with broader implications on the emergence of multi-drug resistance in Mycobacterium tuberculosis.
---

## [Editor Report · Acceptance letter]

16 Nov 2020

Dear Dr. Fortune,

We are delighted to inform you that your manuscript, "Mutations in dnaA and a cryptic interaction site increase drug resistance in Mycobacterium tuberculosis," has been formally accepted for publication in PLOS Pathogens.

Best regards,

Kasturi Haldar

Editor-in-Chief

PLOS Pathogens

orcid.org/0000-0001-5065-158X

Michael Malim

Editor-in-Chief

PLOS Pathogens

orcid.org/0000-0002-7699-2064